# The coupling between hydrology, the development of the active layer and the chemical signature of surface water in a periglacial catchment in West Greenland

Johan Rydberg[1]*, Emma Lindborg[2, 3], Christian Bonde[4], Benjamin M. C. Fischer[5], Tobias Lindborg[6, 3], Ylva Sjöberg[7, 4, 1]

[1]Department of Environmental Science, Umeå University, Linnaeus väg 6, 901 87 Umeå, Sweden
[2]DHI Sweden AB, Svartmangatan 18, 111 29 Stockholm, Sweden
[3]Blackthorne Science AB, Slånbärstigen 36, 125 56 Älvsjö, Sweden
[4]Department of Geosciences and Natural Resource Management, and Center for Permafrost, University of Copenhagen, Øster Voldgade 10, 1350 København K, Denmark
[5]Department of Earth Sciences, Uppsala University, Villavägen 16, 752 36 Uppsala, Sweden
[6]Swedish Nuclear Fuel and Waste Management Company (SKB), Box 3091, 169 03 Solna, Sweden
[7]Department of Physical Geography and the Bolin Centre for Climate Research, Stockholm University, 106 91 Stockholm, Sweden

*Correspondence to*: Johan Rydberg (johan.rydberg@umu.se)

**Abstract.** The chemical signature of surface water is influenced by the interactions with soil particles and groundwater. In permafrost landscapes, ground ice restricts groundwater flow, which implies a limited influence of processes such as weathering on the chemical signature of the runoff. The aim of this study was to examine how freeze-thaw processes, hydrology and water age interact to shape the chemical and stable hydrogen and oxygen isotopic signature of surface water in a catchment in West Greenland. Measuring runoff in remote catchments is challenging, and therefore we used a validated hydrological model to estimate daily runoff over multiple years. We also applied a particle tracking simulation to determine groundwater ages, and used data on stable isotopic and chemical composition from various water types – including surface water, groundwater, lake water and precipitation– spanning from early snowmelt to late in the hydrological active season. Our results show that groundwater is generally younger than one year and rarely exceeds four years. Overland flow is restricted to the snowmelt period and after heavy rain events, while runoff is dominated by groundwater. Monitoring of thaw rates in the active layer indicates a rapid thawing in connection with running water, and meltwater from ground ice quickly becomes an important fraction of the runoff. Taken together, our data suggest that even in continuous permafrost landscapes with thin active layers and an absence of truly old mobile groundwater, soil processes exert a strong influence on the chemical and stable isotopic signature of runoff, similar to what has been observed in other climatic settings.

## 1 Introduction

The water that falls as precipitation carries a signature that reflects the chemical conditions in the atmosphere. This chemical signature – or water quality – is subsequently altered as the water flows through a catchment, where it interacts with water stored in the landscape, vegetation, soil particles and bedrock (Sprenger et al., 2019). The resulting water quality not only determines the ecological and chemical status of local and downstream aquatic systems (EU, 2002), it can also shine light on hydrological pathways and biogeochemical cycling within catchments (Fischer et al., 2015; Lidman et al., 2014; Lyon et al., 2010a; Jutebring Sterte et al., 2022). Knowledge regarding the interplay between soil processes and water also helps us understand how future changes in, e.g., the climate, might affect hydrological pathways and the biogeochemical cycling of elements (Frey and McClelland, 2009; Vonk et al., 2015; Winnick et al., 2017).

Water transit time through a catchment depend on several things, including depth of the regolith and groundwater stores, and water transit times range from days in the shallow layers (0–10 cm) to several hundreds of years as depth increases (Condon et al., 2020). In addition to storage characteristics, the long and cold winters in high latitude regions restrict the flow of water during a considerable part of the year, resulting in the snowmelt period being the single most important event of the hydrological year (Bring et al., 2016; Johansson et al., 2015a). Not only is the runoff high, but the meltwater also flows through a system where ground ice limits the interaction between the meltwater and the soil particles (Bosson et al., 2013; Johansson et al., 2015a). That is, at least during the initial phase of the snowmelt period the water that reaches a stream can be expected to mostly reflect the chemical signature of the melting snow (Chiasson-Poirier et al., 2020).

As the upper soil layers thaw, meltwater from snow can infiltrate the ground surface and interact with decomposing plant material in the organic horizon and mineral soil particles in deeper layers (Cai et al., 2008; Quinton and Pomeroy, 2006). That the upper part of the ground thaws also means that recent snowmelt water can interact and mix with older water that was stored in the ground prior to the onset of the snowmelt event (i.e., pre-event water; Tetzlaff et al., 2018). A prerequisite for the interaction between snowmelt water and soil particles is that the ground thaws while the snowmelt period is still ongoing. This is the case in forested catchments without permafrost in the boreal region, where stream-water dissolved organic carbon (DOC) concentrations increase during the snowmelt period (Jutebring Sterte et al., 2018; Laudon et al., 2004). However, in adjacent wetland catchments where the ice-rich peat results in slower thaw rates, the limited infiltration of snowmelt water results in a decrease in DOC during the snowmelt period and shorter transit time of the water (Jutebring Sterte et al., 2018; Laudon et al., 2004; Lyon et al., 2010b).

In permafrost regions the groundwater storage can be expected to be smaller because the upper part of the ground that thaws each summer, i.e., the active layer, constitutes a much thinner aquifer than what is common in boreal systems without permafrost (Petrone et al., 2016). A result of this can be seen in a study of a polygonal tundra site in Alaska (Throckmorton et al., 2016), where no traces of snowmelt water were found in soil water collected during summer. Similarly, in a catchment-scale study conducted in the Northwest Territories in Canada, Tetzlaff et al. (2018) reported that during the snowmelt period, streamflow was primarily governed by input from snowmelt, and the transit times were short. Over the course of the summer period the transit times then progressively increased to longer than 1.5 years, with a significant increase in the contribution of water from the subsurface system and riparian zones. This suggests that, during the snowmelt period, the presence of impermeable ground ice and slow thaw rates of this ice results in most of the meltwater leaving the system as

overland flow or evaporation. In time, however, deeper flow paths are activated and older pre-event water that
was stored prior to the onset of the snowmelt becomes more important for the chemical signature of the runoff.
As indicated above, thaw rates in the active layer are not only a consequence of air temperatures. On the one hand,
the latent heat of ground ice also plays an important role, which implies that wetter soils can be expected to thaw
slower (Clayton et al., 2021). On the other hand, wet soils have a higher thermal conductivity and could thereby
thaw quicker (Clayton et al., 2021). In addition, running water contains considerable amounts of heat that could
influence the thaw process (Sjöberg et al., 2016). This advective heat transfer suggests that wetter areas and areas
in close connection to surface-water flow paths could thaw faster than drier areas further away from running water.
In addition, microtopography influences the distribution of snow, and because snow insulates the ground during
winter, it can be expected that low-lying areas will have higher ground temperatures in spring (O'Connor et al.,
2019). Taken together this means that it can be difficult to predict how thaw rates and the thickness of the thawed
layer vary spatially across the landscape.
After the intense snowmelt period follows a period with less water moving through the catchment and a
progressively thicker thawed layer. In boreal systems without permafrost, deeper flow paths and increased water
age generally lead to a chemical signature more influenced by soil processes (e.g., weathering and decomposition
of organic matter (OM); Jutebring Sterte et al., 2021a). Higher temperatures also lead to increased evaporation,
which concentrates both elements supplied with atmospheric deposition and those elements supplied through
weathering and decomposition (Alvarez et al., 2015). Taken together we could expect that water collected during
the summer, when transit times are longer, should have a very different chemical signature compared to the
snowmelt period. However, in areas with continuous permafrost the active layer can be less than a meter even
when fully developed in late summer (Petrone et al., 2016). That is, even the deepest flow paths are very shallow
compared to boreal systems without permafrost (Jutebring Sterte et al., 2021b). This implies that the average age
of the runoff water in an area with continuous permafrost is younger, that the flow paths are shorter and that most
of the water leaving the system has had a more limited time to interact with the soil particles compared to boreal
areas without permafrost (Tetzlaff et al., 2018). Hence, the question remains to what extent the water chemistry
in areas with continuous permafrost is affected by – and reflects – processes in the catchment, or if the water
leaves the system too rapidly to pick up any discernible signature from the catchment.
This knowledge is essential to make reliable predictions regarding how future changes in the climate will affect
biogeochemical cycling, especially in Arctic areas with continuous permafrost. However, despite being located
in a global climate hotspot (Rantanen et al., 2022) the processes influencing the water quantity and quality in the
Arctic remain poorly understood, partly because of a declining research infrastructure (Laudon et al., 2017), and
a limited availability of spatiotemporal data due to the region's remote location and extreme climatic conditions
(Tetzlaff et al., 2018; Throckmorton et al., 2016).
In order to better understand how flow paths, water age and chemical signatures covary over different time scales
in a periglacial landscape with continuous permafrost we have used a well-studied catchment in West Greenland,
i.e., the Two-Boat Lake catchment (Johansson et al., 2015b; Lindborg et al., 2016; Lindborg et al., 2020; Petrone
et al., 2016; Rydberg et al., 2023; Rydberg et al., 2016). A key novelty of the present study is the coupling of field
observations of water chemistry and thaw rates in the active layer with a well-constrained, distributed,
hydrological model (Johansson et al., 2015a). This approach allows us to address the following specific research
questions: i) can we observe any relation between soil wetness and the thaw rate in the active layer during the
snowmelt period, ii) does the chemical signature and stable isotopic composition of the runoff suggest that the
thaw in the active layer is fast enough to allow the meltwater from snow to interact with soil particles and pre-
event water during the snowmelt period, iii) considering the dry conditions and the thin aquifer, is the interaction
with soil particles the main factor in driving the chemical composition during the unfrozen season, or do our
samples suggest that there are other important factors that contribute to shaping the chemical signature of the
water.
**2 Methods**
**2.1 Study area**
Two-Boat Lake (also referred to as SS903, lake area: 0.37 km$^2$) and its terrestrial catchment (1.7 km$^2$) are situated
in West Greenland (Lat 67.126° Long -50.180°), about 25 km east of the settlement of Kangerlussuaq (Fig. 1).
Even though the Greenland ice sheet is only about 500 m from the lake, it receives no direct input of glacial
meltwater from the ice sheet (Johansson et al., 2015a). Permafrost in the Two-Boat Lake catchment is continuous
and reaches down to about 400 m, except under the lake itself where a through talik has formed (Claesson Liljedahl
et al., 2016). Bedrock consists mostly of tonalitic and granodioritic gneisses (Van Gool et al., 1996) and is covered
by till or glaciofluvial deposits that are, in turn, overlain by a layer of eolian material (Petrone et al., 2016). The
climate of this region is cold and dry, with a mean annual air temperature for the Two-Boat Lake catchment of -
4.4 °C (Johansson et al., 2015b). The annual precipitation for the period 2011-2013 was 270 mm yr$^{-1}$ (163-366
mm yr$^{-1}$) with about 40% falling as snow. The high evapotranspiration in the terrestrial part of the catchment (138-
199 mm yr$^{-1}$) results in the runoff to the lake is relatively low (25-159 mm yr$^{-1}$) and most of the runoff is associated
with the snowmelt period (Johansson et al., 2015a). There are no permanent streams or confined stream channels
in the catchment. Instead, surface runoff occurs in small (typically <0.5m wide), temporary surface streams that
are active only during high runoff situations, particularly the snowmelt period (Johansson et al., 2015a). The
vegetation zone in the area can be characterised as polar tundra or steppe (Bush et al., 2017; Böcher, 1949), and
the vegetation of the Two-Boat Lake catchment has been described in detail by Clarhäll (2011). Between 2010
and 2019 the catchment was the study site of the Greenland Analogue Surface Project (GRASP), which was
funded by the Swedish Nuclear Fuel and Waste Management Company (SKB). During GRASP, data regarding
meteorology, hydrology, Quaternary deposits, vegetation cover and active layer thickness were collected together
with samples from soils, groundwater, surface- and lake water for chemical analysis. For this study we have
focused on a sub-catchment (0.6 km$^2$) in the northern part of the Two-Boat Lake catchment (Fig. 1).

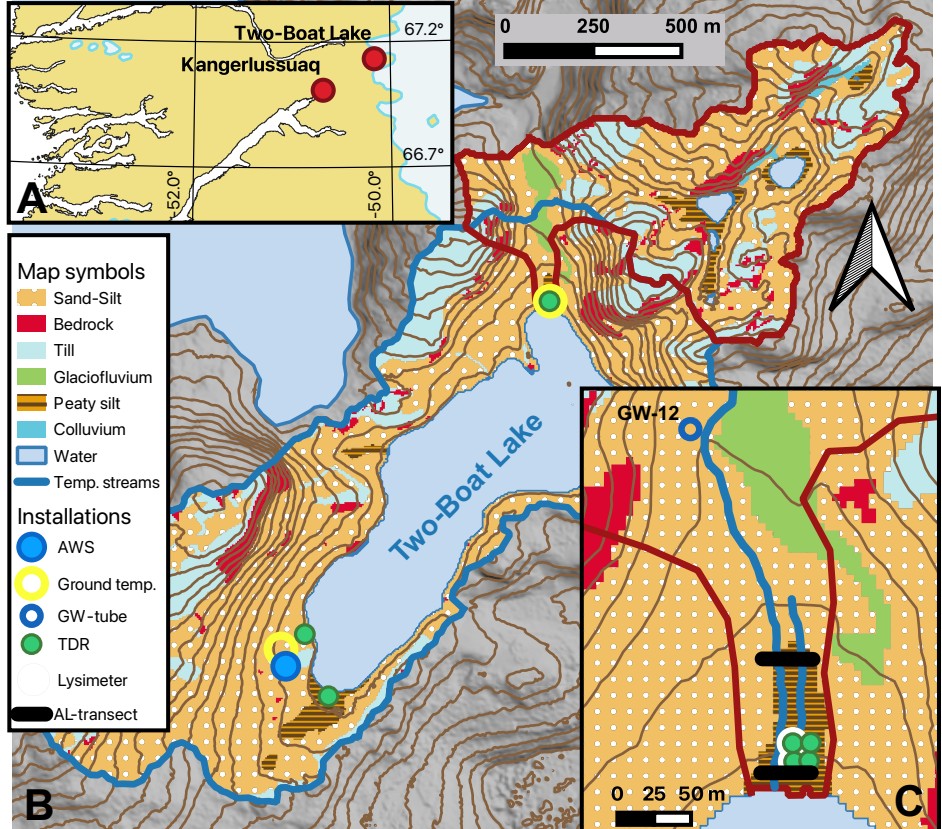


**Figure 1: Map showing the location of the Two-Boat Lake catchment within West Greenland (panel A). Panel B shows a regolith map of the entire Two-Boat Lake catchment (peaty silt corresponds to wetter areas), including the location of the automated weather station (AWS), ground temperature loggers, and TDR-installations. The red outline corresponds to the sub-catchment used for this study, while the blue outline marks the original catchment area used by Johansson et al. (2015a). Panel C shows a close-up of the lower part of the studied sub-catchment, including the two AL-transects that were used to monitor the thaw in the active layer during spring 2017 and the two temporary streams that were active during this period. It also shows the location of the groundwater well (GW-12) and two lysimeter installations that were used as observation points in the particle tracking simulation, as well as the four TDR depth profiles. The digital elevation model used to create the contour lines (10-m contour intervals) and the regolith map were developed within the GRASP-project and are described in detail in Petrone et al. (2016).**

## 2.2 Modelling

### 2.2.1 Hydrological modelling

During GRASP, Johansson et al. (2015a) used the modelling system MIKE SHE to develop a distributed numerical hydrological model for the entire Two-Boat Lake catchment. Here we have extended that model to cover the period 2011-2019. The model uses meteorological data collected by an automated weather station (AWS) in the Two-Boat Lake catchment (Fig. 1) and has been thoroughly validated using on-site measurements of several hydrologic variables (see Johansson et al. 2015a for details on model setup and validation). The model domain has a horizontal resolution of 10x10 m, and covers the entire catchment, as well as the area directly downstream of the lake outlet (to avoid boundary effects). Vertically, the model is composed of four 25-cm thick layers in the seasonally thawed active layer (top 100 cm), where water flow occurs seasonally, and four layers in the underlying permafrost (1-200 m depth), where water flow is extremely low due to the frozen conditions.

Observed daily precipitation and potential evapotranspiration (calculated from observed meteorological data) are applied as the upper boundary condition, and the model simulates the catchment water balance by partitioning the incoming precipitation into evapotranspiration, surface runoff and/or infiltration for each cell in the model. The subsurface is composed of an unsaturated zone from which water can evaporate via the soil matrix or transpire through plant roots, percolate to the saturated zone, or remain in storage. The unsaturated zone varies in thickness depending on the depth of the simulated groundwater table. In the saturated zone, below the groundwater table, water can move vertically and horizontally as simulated by Darcy's law.

Because MIKE SHE does not itself simulate freeze-thaw processes (i.e., permafrost), the impact of freeze-thaw processes on the hydrological flow in the active layer and permafrost was parameterized by spatially and temporally varying the hydraulic properties (hydraulic conductivities, surface roughness and infiltration capacity) of the ground substrates based on the observed ground temperatures at different depths and locations in the catchment. By using this approach, MIKE SHE handles permafrost by simulating the hydrological consequences of frozen and thawed conditions rather than simulating ground temperatures (Bosson et al., 2013). That is, water mobility in the subsurface zones is effectively restricted when the ground is frozen. For more details regarding the model setup and validation, please refer to Johansson et al. (2015a, 2015b).

Code developments in MIKE SHE between the original model and the extended model run for the present study resulted in two model versions and associated evaluation periods. For the original 2010-2013 period, we used the output from the original modelling run made by Johansson et al. (2015a). For the 2013-2019 period the same model was used, but with the inclusion of an additional area. This additional area only contributes runoff to the lake during high flow situations, e.g., during the snowmelt period or after periods with intense rain, and it is not connected to the rest of the catchment via a predefined stream network in the model. When ponded water accumulates in this area, it is routed downhill as overland flow or infiltrated to reach downstream areas of the catchment as shallow groundwater flow.

### 2.2.2 Particle tracking simulation to estimate groundwater age

In order to estimate the age of the water in the saturated zone of the studied sub-catchment we used the particle tracking module in MIKE SHE together with the numerical hydrological model described above (Fig. 1). This module adds virtual "particles" to the water that enters the saturated zone of the hydrological model. These particles then move (conservatively) with the water through the saturated zone until they leave the saturated zone into the lake, a temporary surface stream, standing surface water, the unsaturated zone or a model boundary. Based on the time when they entered the saturated zone and when they reached one of the three observation points it is possible to get an estimate of the age of the groundwater (Fig. 1).

The MIKE SHE particle-tracking module releases particles along the established 3D flow field in the saturated zone only. The number of particles released was flow-weighted, i.e., the more water that enters the saturated zone during a certain period, the more particles were released during that period. In this case one particle was released for each millimetre of added water. Once a particle leaves the saturated zone, the particle is removed from the simulation. To ensure enough time for most particles to leave the saturated zone or reach the observation points, particles were only released during the first year of the 100-year simulation and the hydrological model (i.e., the MIKE SHE hydrological model described above) was cycled to create a 100-year simulation using the 2016-2019 meteorological conditions. Groundwater age, i.e., the transit time in the saturated zone, was then estimated by

noting the date the particle entered the saturated zone and the date when the particle reached an observation point (i.e., one of the groundwater wells (GW-12) and the two lysimeter installations found within the sub-catchment; Fig. 1). Each observation point was subdivided into three vertical layers in accordance with the hydrological model (i.e., 0–25 cm, 25–50 cm, 50–75 cm).

## 2.3 Sample collections and field measurements

### 2.3.1 Monitoring of thaw rates in the active layer

Between May 18th and May 31st 2017, the progress of thaw in the active layer was monitored using a steel rod along two 50-m long transects (AL-transects; Fig. 1). Every second day (7 times in total) we measured the thickness of the thawed layer in 0.5-m intervals along the transects. Each measurement spot was also classified as wet (water table at or above the ground surface) or dry (water table more than 5 cm below the ground surface), and we measured the depth of any surface water. Thaw rates were then calculated by dividing the change in thaw depth between two consecutive monitoring dates by the time between the two dates. When measuring the thaw depth care was taken not to walk on the measured transect and we avoided walking along the same path twice. No visible signs of tracks or rilling were observed either during the intense monitoring period or during the entire GRASP study period. Thaw rates for the whole monitoring period and for each spot along the transects were calculated as the average thaw rate for the entire period.

### 2.3.2 Meteorology and ground temperatures

Air temperature and precipitation (as both rain and snow) were recorded using an automated weather station (AWS) placed in the Two-Boat Lake catchment, and ground temperatures were measured at two locations in the catchment using HOBO U12-008 sensors (Fig. 1; see Johansson et al., 2015b for further details). Time Domain Reflectometry (TDR) sensors (Campbell Scientific, CS615 sensors connected to a CR1000 data logger) were placed in three clusters, each consisting of either three or four depth profiles with four sensors evenly distributed from 5-10 cm below the ground surface down to 40-50 cm (Fig. 1; see Johansson et al., 2015b for further details).

### 2.3.3 Estimate of the amount of ground ice

To estimate the amount of water released from melting ground ice during the intense monitoring period of 2017, we used the soil moisture content recorded by the TDR sensors in the sub-catchment at the time of freeze-up in the fall of 2016. These data allowed us to estimate the ground ice content at different depths, and thus, estimate the release of meltwater as the thaw progresses. This estimate should be considered as a minimum, because moisture migration towards the thaw front could give a higher ice content in spring compared to the fall conditions. The amount of water released from melting ground ice between May 18th and May 31st, 2017, was estimated based on the water content linearly interpolated between observation depths (TDR sensors) and the maximum thaw depth for each point along the transects at the end of the intense monitoring period.

### 2.3.4 Collection of water samples and chemical analysis

Precipitation (rain and snow; n=8 and n=13, respectively), surface-water (n=31), groundwater (groundwater wells and zero-tension lysimeters, n=23 and n=22, respectively) and lake-water (n=28) samples have been collected at

an irregular, low, frequency between 2010 and 2019 (Lindborg et al., 2016). Groundwater wells are fully screened,
and collects water from the entire thawed layer, while lysimeters collect water at discrete depths (i.e., either 15 or
30/35 cm depth). The ceramic body of the lysimeters has a pore size of 1 μm and no additional filtering was done.
For other sample types, the majority were filtered in the field using 0.45 μm polycarbonate membrane filters (two
lake water and two surface water samples were analysed unfiltered). All samples were frozen after sampling and
were kept frozen until analysis. Water samples have been analysed for their elemental composition (n=129) using
Inductively-Coupled Plasma Sector-Field Mass Spectrometry (ICP-SFMS) at ALS in Luleå, and/or dissolved
organic carbon (DOC; n=62) using the NPOC-method at either Stockholm University (Dept. of Ecology,
Environment and Plant Science), ALS in Luleå or Umeå University (Dept. of Ecology and Environmental
Science). Please refer to Lindborg et al. (2016) for further details.
In addition to this low-frequency sampling, a number of samples were collected simultaneously with the intense
monitoring period (i.e., between May 18th and May 31st, 2017). One set of samples was used for ICP-SFMS and
DOC measurements. This sample set consists of samples from where one of the temporary surface streams enters
Two-Boat Lake, the uppermost groundwater well (GW-12), one lysimeter (15-cm depth) and the snowpack above
the groundwater well (Fig. 1). A second set of samples was used for analysing stable water isotopic signatures
($\delta D$ and $\delta^{18}O$). Surface water was sampled every second day from two small temporary surface streams that
crossed the transects used to monitor the thaw depth. Snow was sampled from three locations in the sub-
catchment, and water from ground ice in the active layer was sampled in three locations along one of the temporary
surface streams by thawing pieces of frozen ground in plastic bags and sampling the meltwater. All samples were
taken as duplicate samples (field replicates), and the stable isotopic composition in each sample was analysed in
triplicate (laboratory replicates). Stable water isotopes in all water samples from the 2017 sampling were analysed
at Stockholm University using a Cavity Ring-Down Spectrometer (Picarro L2130-I, manufacturer's precision of
$\delta D < 0.1$‰ and $\delta 18O < 0.02$‰). The analysis scheme of Penna et al. (Penna et al., 2010) was adopted, and results
were reported as $\delta$-values in per mille (‰) relative to Vienna Standard Mean Ocean Water (VSMOW). All
calculations and statistical treatments were made using the average stable isotopic composition of the laboratory
replicates.

**2.4 Data analysis**

**2.4.1 Principal component analysis**

To evaluate and visualize similarities and differences between different types of water samples the chemical data
from rain, snow, temporary surface streams, groundwater wells, lysimeters and the lake itself were subjected to a
principal component analysis (PCA). For temporary surface streams, groundwater wells and lysimeters only
samples from the sub-catchment were included. Prior to the PCA, elements for which the majority of observations
were below the reporting limit in all compartments were removed. For the remaining elements all values below
the reporting limit were replaced with half the reporting limit, and the dataset was then converted to z-scores
(average=0; variance=1) to remove any effects of scaling. After an initial PCA, we also removed elements with
communalities <0.7, as well as two snow samples that had elemental concentrations similar to lake water
indicating that they had been contaminated by lake water (collected 2014-04-11) and a single sample from a
groundwater well (GW-11, 2011-09-13) that had an disproportionally strong influence on the outcome of the
PCA. This resulted in a total of 35 elements and 65 observations (rain=8, snow=6, surface water=7,
groundwater=23, lake water=11). All principal components (PCs) with eigenvalues above one were extracted,
and a Varimax rotated solution was used.

### 2.4.2 Correlation analysis

All statistical calculations for the comparisons between hydrological variables and the chemical signature were
performed using SPSS v.29 (www.IBM.com). Correlation coefficients were calculated as Pearson correlations
(denoted $r$) if both variables were normally distributed (according to a Shapiro-Wilk test) or Spearman rank
correlations (denoted $r_s$) if at least one variable was non-normally distributed. For all tests, a significance level of
0.05 was used. The majority of the runoff in the Two-Boat Lake catchment occurs as groundwater, and several of
the hydrological parameters reported from the model are highly correlated. Therefore, only the total runoff to the
lake and the proportion of deep groundwater were used for the correlation analysis (these two parameters show
no correlation to each other).

### 3 Results

### 3.1 Hydrological modelling,2013-2019

The observed average annual precipitation for the extended modelling period (i.e., 2013-2019) was 308 mm yr$^{-1}$,
and varied from a minimum of 247 mm yr$^{-1}$ in 2014 to a maximum of 407 mm yr$^{-1}$ in 2017. Based on the local
meteorological observations, the hydrological model estimates that, on average, two thirds of the precipitation left
the system as evapotranspiration (65%), and the period from late April through October was characterized by a
precipitation deficit (i.e., precipitation minus evapotranspiration was less than zero), whereas the winter period
had a precipitation surplus. On an annual basis the modelled average net input of water to the catchment (i.e.,
precipitation minus evapotranspiration) was 109 mm yr$^{-1}$, with a variation from 64 mm yr$^{-1}$ (2018) to 170 mm yr$^{-1}$
$^{1}$ (2017). Viewed over the 2011-2019 period most of this excess water left the terrestrial system as runoff (on
average 98 mm yr$^{-1}$) and entered the lake. The snowmelt period was the dominant runoff event in all modelled
years and most commonly the runoff peaked in early June (Fig. 2). During the early snowmelt period a substantial
part of the runoff occurred as overland flow directly to the lake (i.e., water that had not been in contact with the
subsurface), but for the latter part of the snowmelt period and the summer season the runoff was dominated by
groundwater discharging directly to the lake or via temporary surface streams (Fig. 2). The exception to this
pattern was 2018, when overland flow also occurred late in the snowmelt period and during summer. On an annual
basis overland flow made up 7-52% of the annual runoff (average 28%), with 2016 and 2018 standing out from
the other years with about half the water leaving the catchment as overland flow.
Observed ground temperatures down to two meters depth showed a clear seasonal pattern that became muted with
depth (Fig. 2). On average for the 2011-2018 period the maximum thaw depth, which occurred in August, reached
the temperature sensor at 0.75 m. The thaw period at 25 cm started in late May (May 29th) and reached 50 cm by
mid-June (June 20th). The ground then stayed unfrozen for about four months, and froze at 25 cm in late September
and in mid-October at 50 cm. That the deeper parts of the active layer (at 50 cm) stayed unfrozen for almost a
month after air temperatures dropped below zero degrees and the surface layers froze means that groundwater
flow could continue also after the ground surface had frozen.

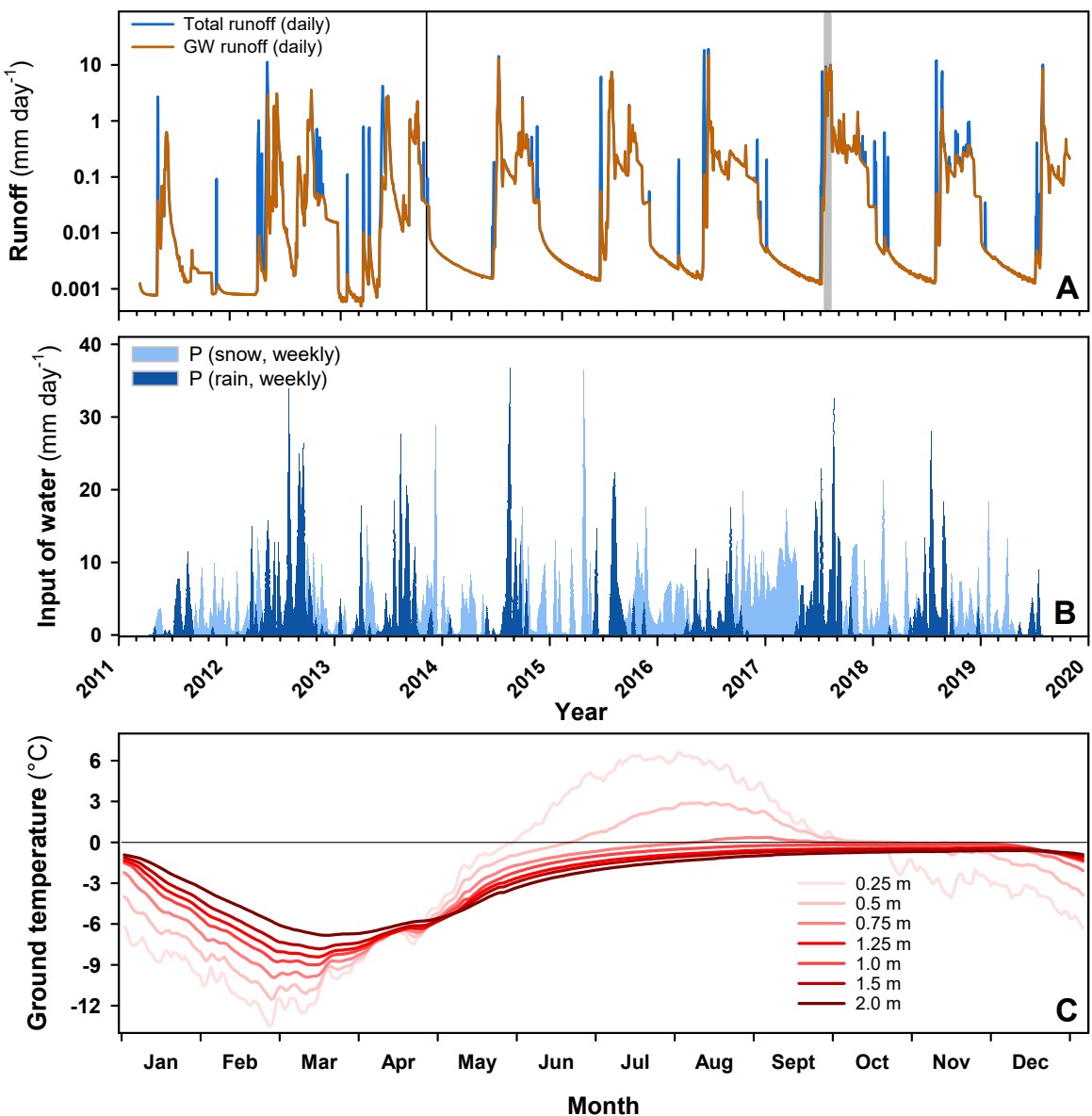

**Figure 2. Panel A shows the modelled average daily runoff (total runoff and as groundwater) to Two-Boat Lake for the**
**entire modelling period (March 2011 to July 2019). The black vertical line denotes the division between the original**
**modelling run presented in Johansson et al. (2015a), and the extended period (2013-2019). The shaded grey area**
**indicates the intense monitoring period in 2017. Panel B presents the observed weekly precipitation as snow (air**
**temperature was below zero) or rain for the Two-Boat Lake catchment. The lower panel (C) shows the observed ground**
**temperature (daily average for 2011-2018) measured down to 2 m depth (cf. Fig. 1 for location)**

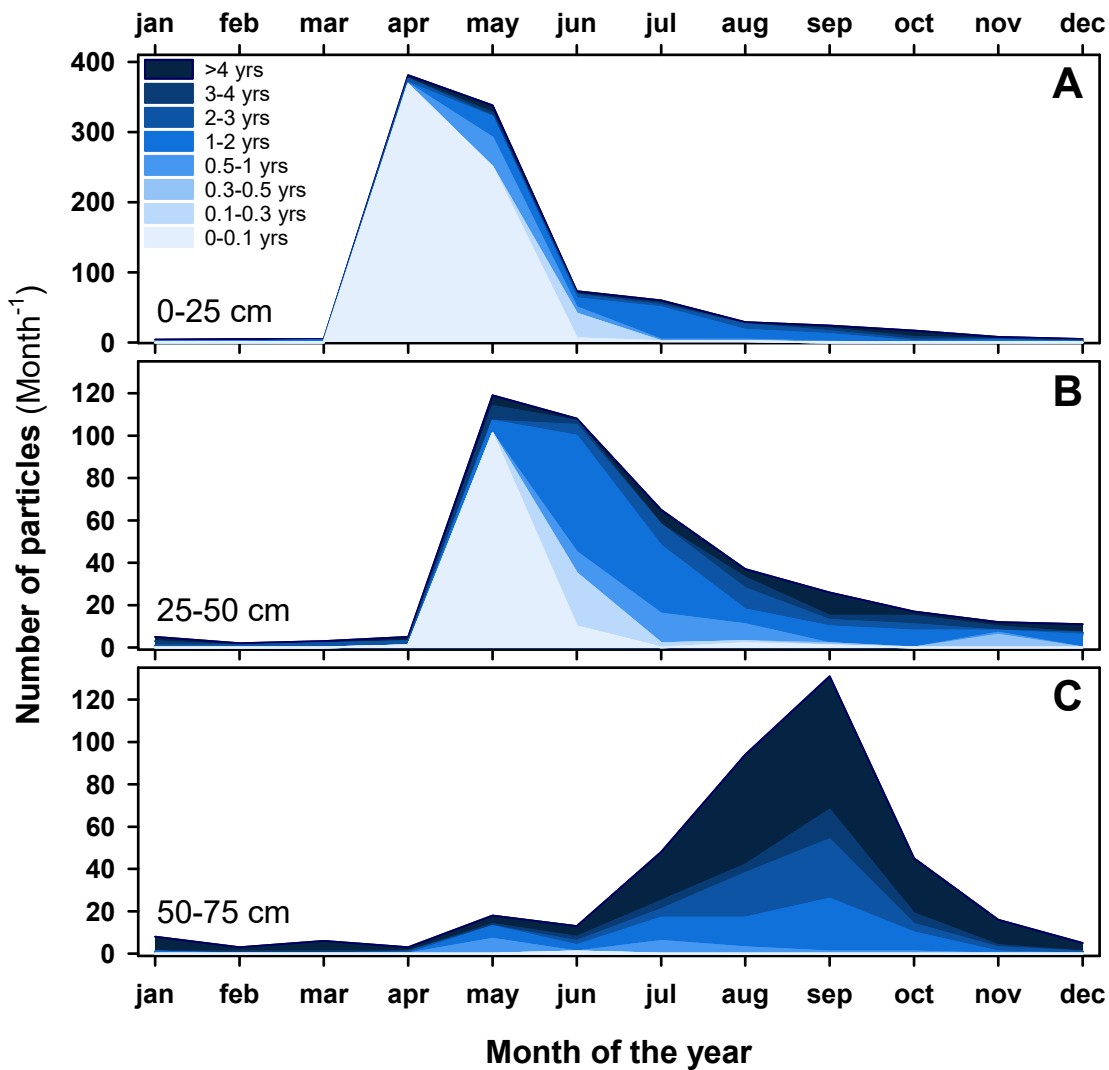


**Figure 3, Age distribution for water particles reaching the three observation points used in the particle tracking**
**simulation for three depth layers (A: 0-25 cm, B: 25-50 cm and C: 50-75 cm). Note that the scale differs between panel**
**A and panels B and C.**

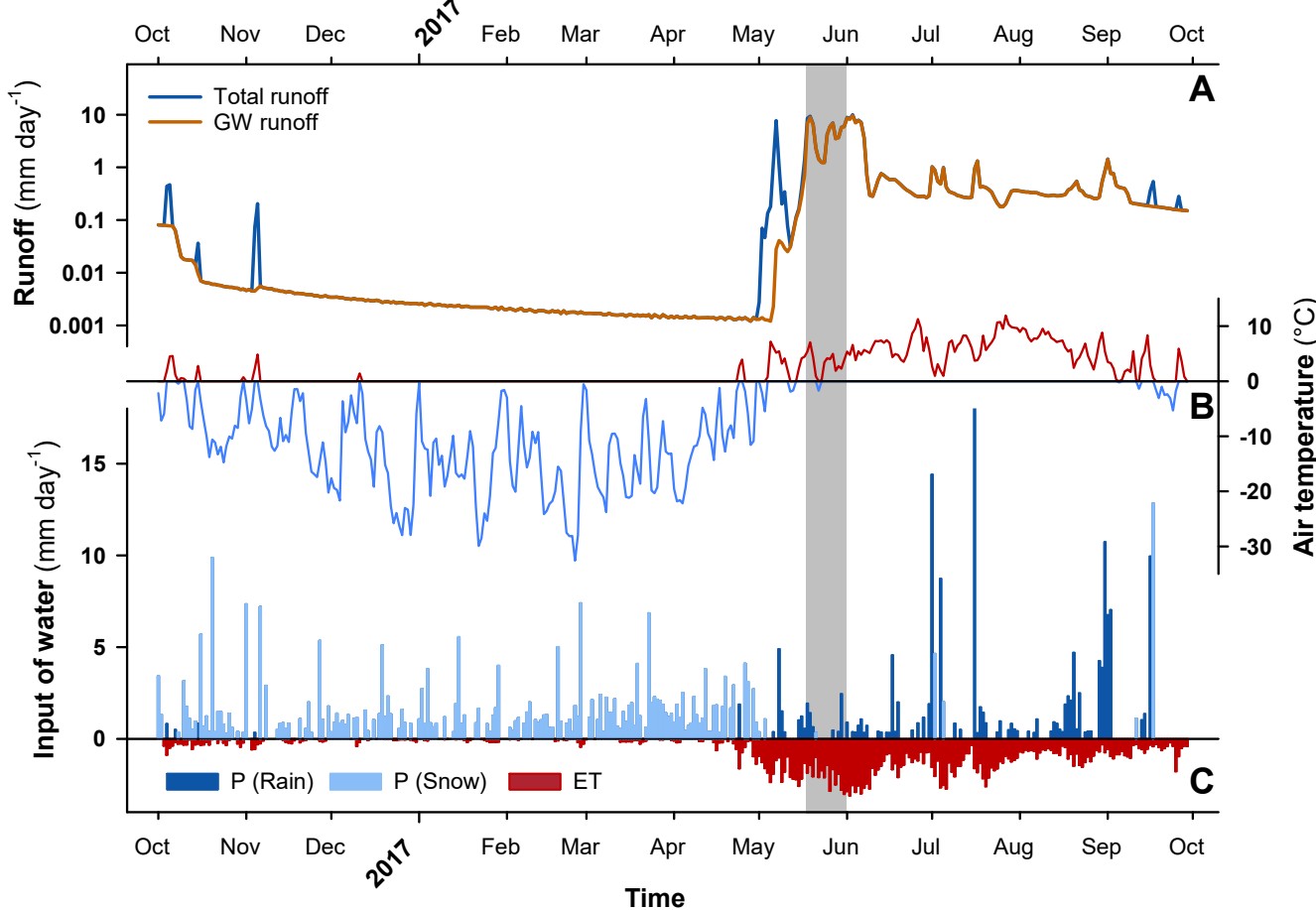

**Figure 4. Modelled average daily runoff from the hydrological model for the period 2016-10-01 to 2017-09-30 (A), observed daily air temperature (mean) for the Two-Boat Lake catchment for the same period (B), and precipitation as snow or rain for the same period (C). The intense monitoring period (May 18th to May 31st) is indicated by the shaded area.**

### 3.2 Particle tracking simulation and groundwater age

The particle tracking simulation suggests that most of the tracked particles moved through the uppermost layer of the model (i.e., 0-25 cm; Fig. 3). From the start of the snowmelt period until the end of June–when most particles are registered at the observation points–the 0-25 cm layer was dominated by very young water that generally entered the saturated zone within the previous 3 months. As the unfrozen season progressed, the average age of the tracked particles increased in all three layers, and the flow also shifted to deeper flow paths as the unfrozen season progressed. In deeper layers the peak in the number of registered particles occurred later in the season (May-June and September for 25-50 cm and 50-75 cm, respectively), and the particles registered in the deeper soil layers also tended to have higher ages (Fig. 3). Almost all the particles that moved through the deepest part of the active layer had spent more than a year in the saturated zone, and about two-thirds of the particles were older than two years. It should be noted that the ages only relate to particles released as water enters the saturated zone, hence, it is not possible to track the age of water that was present in the saturated zone at the start of the simulation. Also, because the 100-year simulation does not include any warming trend (it repeatedly cycles the 2016-2019 meteorological conditions), it does not include any release of water from thawing permafrost.

**3.3 Intense monitoring period in spring 2017**

During the winter 2016/2017, which preceded the intense monitoring period in late May 2017, the precipitation AWS in the Two-Boat Lake catchment indicates that 260 mm of snow-water equivalent accumulated in the catchment. According to the hydrological model 26.5 mm of this water was lost due to sublimation or evaporation (Fig. 4). Based on the measured air temperature the hydrological model indicates that the snowmelt started on April 30th, with about 8% of the snowmelt-associated runoff occurring before May 13th, mainly as overland flow directly to the lake (Fig. 4). This means that a substantial part of the accumulated snowpack had already melted (or sublimated) at the beginning of the intense monitoring period (May 18th to May 31st), and snow patches were mainly confined to higher elevations in the catchment. Even so, multiple small temporary surface streams were still observed across the catchment, and according to the hydrological model the runoff remained high until June 10th, but during the latter period most of the water had spent some time as groundwater before entering a temporary surface stream or reaching the lake.

Time Domain Reflectometry (TDR) sensors installed in the sub-catchment showed a soil moisture content of approximately 30% during soil freeze-up in the fall of 2016. Furthermore, the ground temperature sensors in the sub-catchment showed that thawing of the ground at 5-cm depth started on May 10th, and reached 15-cm depth on May 20th. The melting of ground ice in the active layer corresponds to a total water release of at least 18 mm of water during the sampling period, assuming that the recorded soil temperatures and water content at the TDR sensors are representative of the entire sub-catchment. During the intense monitoring period (May 18th to May 31st) there was also an input of a total of 7 mm of rain, with the largest rainfall event of 0.8 mm occurring on May 30th.

**3.4 Thaw rates in the active layer**

The average thaw depth along the two AL-transects closely resembled the thaw depth observed using the ground temperature sensors in the sub-catchment, but there was considerable variability in ground thaw along the two transects (Fig. 1). At the start of the intense monitoring period (May 18th) the average thaw depth along the studied transects was 10.4 cm, and by the end of the period (May 31st) it had increased to 15.3 cm. On the first day of sampling, May 18th, 30% of the measurement points had a water level at or above the ground surface, and 7% were covered by surface water. The surface water formed several puddles, and two temporary surface streams that crossed at least one of the AL-transects (Fig. 1). During the monitored period wet locations (with water table at or above the ground surface) exhibited a higher thaw rate (0.8 cm day$^{-1}$) than dry locations (0.6 cm day$^{-1}$, p-value <0.0001). The maximum observed thaw occurred under one of the temporary surface streams, where the thawed layer reached down to 48 cm below the ground surface on the last day of the monitoring period (May 31st).

**3.5 Stable water isotopes**

The stable water isotopic composition of water from temporary surface streams sampled in May 2017, as well as the average for precipitation and lake water taken in 2011, 2014, 2017, and 2019, are shown in Fig. 5 and SI-Fig. 1. For the precipitation there was a clear seasonality in the composition with more negative values in the snow samples (average $\delta D$ = -159.7 ‰, standard deviation 12.5 ‰) compared to rain samples (average $\delta D$ = -115.8 ‰, standard deviation 9.3 ‰). The compositions in samples from surface water (average $\delta D$ = -140.6 ‰, standard

deviation 2.0 ‰) and ground ice (average = -137.5 ‰, standard deviation 0.8 ‰) showed a much smaller spread, and the stable water isotopic values fell between the values for snow and rain (Fig. 5).

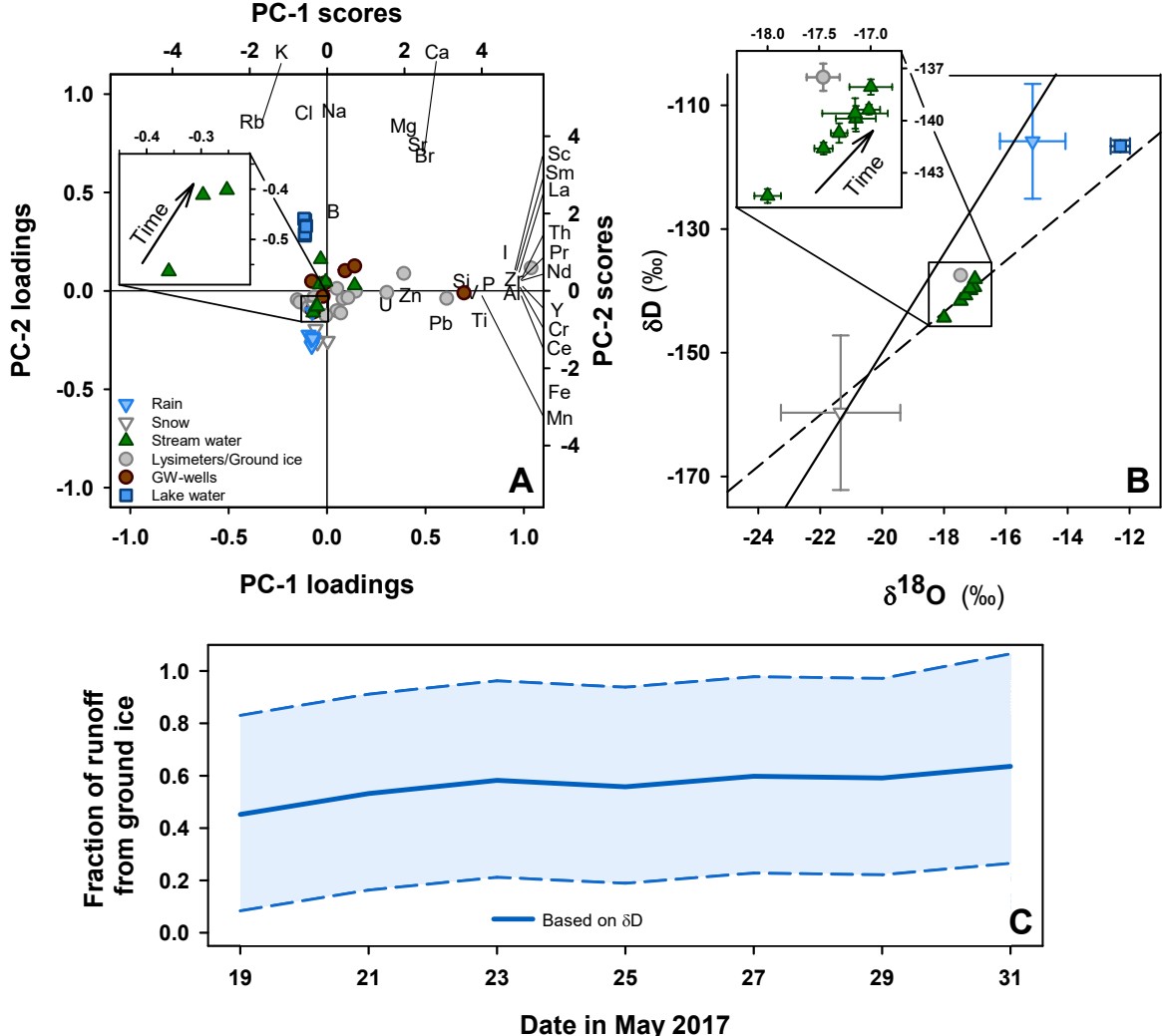

**Figure 5, Panel A consists of a combined loadings and score plot for the first two PCs (PC-1 and PC-2) intended to provide an overview of how the chemical signature varies between water types. Precipitation samples are found in the lower left-hand corner, while lake water samples are found in the upper left-hand corner. Surface water and soil water samples (lysimeter and groundwater wells) generally have similar or higher PC-1 scores compared to precipitation and lake water, with the highest scores found in soil water samples. For PC-2 surface and soil water are intermediate to precipitation and lake water samples. The insert shows the three water chemistry samples from the intense monitoring period, with the arrow indicating the trend with time. Panel B shows the average stable water isotopic composition (δD and δ$^{18}$O) in snow, rain, ground ice samples, surface water and lake water samples. The symbols indicate the average, while the error bars indicate the standard deviation of all samples collected over the entire study period. The solid line represents the GMWL and the dashed line represents an evaporation line to indicate the trajectory of evaporating water. The insert shows the stable isotopic composition in the six surface water samples and the melted ground ice sampled in May 2017. The arrow indicates the trend with time, and error bars represent the variability between replicates. For access to the full datasets for both water chemistry and stable isotopic composition we refer to Lindborg et al. (2016) and its associated Pangaea databases. Panel C shows how the fraction of runoff from melting ground ice**

**and/or rain increases in the surface water over the intense monitoring period. The mixing model is based on the change in δD in the temporary surface stream and using the average of rain (including ground ice) and snow as endmembers.**

During the two-week intense monitoring period the stable isotopic composition in samples from the temporary surface streams changed considerably (Fig. 5). At the end of the monitoring period the δD signature of the surface water was heavier and more closely resembled ground ice and/or rain than snow. The mixing model – which was based on the δD in temporary surface streams, rain (including ground ice), and snow samples – showed that on the first day of sampling 39% of the surface water originated from rain, while 49% came from rain on the last day of sampling (Fig. 5). The uncertainty range is high due to the high spread of values in snow and rain samples. Only 7 mm of rain fell during the intense monitoring period (and 3 mm the week before), while the estimated water released from melting of ground ice in the thawed part of the active layer was 18 mm. Most of the rainwater therefore likely originated from rain that fell during the previous year and that had been stored as ground ice during winter.

### 3.6 Surface water chemistry

The PCA identified five principal components (PCs) with eigenvalues above one, and together they explain 90% of the variance in the data (Fig. 5). The first PC (PC-1) is driven by differences in elements related to silicate minerals that can be assumed to be released through weathering (e.g., Al, Ce, Nd, La, Pr, Sc, Th, Y, Zr, P; Deer et al., 1992). A part of the variability in Mg, Sr and Ca, which also are released through weathering, is also associated with this PC (the elements plot in the upper right-hand quadrant). Samples with high PC-1 scores come from lysimeters and groundwater wells, i.e., water that has had more contact with soil particles, and samples from precipitation and lake water have negative scores. DOC, which could not be included in the PCA because it was not analysed in all of the samples used for ICP-MS analyses, showed a high positive correlation with PC-1 scores ($r_s$=0.95, p-value <0.001). The second PC (PC-2) is driven by elements that originate from the ocean (e.g., Cl and Na), and that are delivered to the catchment with precipitation. In addition, a part of the variability in elements like Ca, Mg, Sr and K – which are all present in the precipitation but that could also be released during interaction with soil particles – is associated with PC-2. For PC-2 there is a split between samples from the lake on the positive side and precipitation on the negative side (with soil water samples in the middle). That the surface and lake water have considerably higher concentrations of elements not released by weathering (e.g., Cl) as compared to the precipitation indicates that an enrichment caused by evaporation (cf. Rydberg et al., 2023). The remaining three PCs are driven primarily by a small number of samples without any obvious trends, and they are therefore difficult to interpret in terms of general processes. For the remainder of this paper, we will focus on PC-1 and PC-2, which both reflect processes that vary over the course of the year, e.g., depending on the source of the water, the air temperature and the intensity in the runoff.

### 3.7 Correlations between hydrology and chemical signature of surface and soil water

During the intense monitoring period samples for geochemical analyses were collected on three occasions (May 19[th], 25[th] and 31[st]). During this period the hydrological modelling suggests that the total runoff as both groundwater and overland flow decreased (from 9.4 mm to 4.2 mm between May 19[th] and May 25[th]), and that it then increased slightly to 6.0 mm on May 31[st]. During the same period the proportion of deep groundwater

increased from 2.5 to 4.1 and finally 4.4% of the total runoff. Looking at the PC-scores, both PC-1 and PC-2
scores increased over the intense monitoring period. The PC-1 scores went from -0.36 on May 19th, to -0.30 and
then -0.25 on May $31^{st}$. For PC-2 the main shift occurred between the $19^{th}$ and $25^{th}$ when the score increased from
-0.56 to -0.41. On May $31^{st}$ the PC-2 score was essentially the same as on May $25^{th}$ (-0.40). During this period
DOC changed from 20.8 mg $L^{-1}$ to 13.5 mg $L^{-1}$ and finally 14.7 mg $L^{-1}$.
If we look at surface-water samples collected in the studied sub-catchment during the entire 2011-2019 period
(n=7) the chemical signature in the surface water varies considerably. When the proportion of deep groundwater
is high, there is an increase in the concentrations of weathering products released from soil particles (i.e., PC-1
scores). This results in a positive correlation between PC-1 and the proportion of deep groundwater ($r_s$=0.79,
p=0.036). For elements where the source is mainly precipitation and that are enriched through evaporation (i.e.,
PC-2), there is an increase in the concentrations as the total runoff decreases over the course of the unfrozen
season. For example, PC-2 scores are positively correlated with day-of-year (DOY; $r_s$=0.89, p=0.007), and there
is a negative correlation between PC-2 scores and total runoff (r=-0.81, p=0.029). For the surface-water samples
neither PC-1 nor PC-2 scores are correlated with DOC, but similar to PC-2 scores, DOC is negatively correlated
with the total runoff (r=-0.59, p=0.017).
When looking at the lysimeters and groundwater wells – i.e., the soil water – a different pattern emerges. First,
the chemical signature in the soil water is even more variable compared to the surface water. Second, PC-1 and
PC-2 are correlated ($r_s$=0.47; p=0.021), and both are correlated with DOC (r=0.92, p=0.001 and r=0.86, p=0.001,
respectively). This indicates that even if the processes driving PC-1 (weathering), PC-2 (evapotranspiration) and
DOC (organic matter decomposition) are not related, there is a tendency that soil water is enriched in all three in
a similar way. In the soil water, DOC shows a negative correlation with total runoff ($r_s$=-0.92, p<0.001), while
neither PC-1 nor PC-2 scores for the soil water are correlated with any of the hydrological variables, and neither
PC-1, PC-2 nor DOC are correlated with sampling depth.
**4 Discussion**
**4.1 Young water and shallow flow paths dominates the runoff**
The output from the extended hydrological modelling period (2013-2019) largely corroborates the findings from
the original modelling period (2011-2013; Johansson et al., 2015a). That is, Two-Boat Lake is situated in a dry
periglacial landscape, where a considerable fraction of the precipitation (55-80%) is lost through sublimation or
evapotranspiration in the terrestrial system, and where surface water runoff is mostly confined to high flow
situations during the snowmelt period. That there are not any major differences in the hydrological situation is in
line with findings that the study area has not experienced any significant change in climate over the study period
(2001-2019), even though a long-term (1981-2019) warming trend has been suggested for the region (Hanna et
al., 2021). All years except 2017, which was slightly wetter (405 mm $yr^{-1}$) compared to the previously wettest
year, i.e., 2012 (366 mm $yr^{-1}$), fall within the range of 2011-2013 for precipitation, evapotranspiration and total
runoff from the catchment to the lake (Fig. 2). The annual pattern in runoff is also very similar. Runoff is highest
during the snowmelt period, which accounts for about half of the annual runoff to the lake (Johansson et al.,
2015a). During summer the total runoff decreases as a response to increased evapotranspiration (which generally

exceeds the input of water during this period), while in the fall (August-September), evaporation decreases and the total runoff to the lake increases again (Fig. 2).

According to our particle tracking simulation most of the water that moves through the sub-catchment is young, especially during the snowmelt period when almost all groundwater is young (i.e., less than 1-year old). The simulation also suggests that old mobile groundwater (i.e., more than 10 years) is lacking in Two-Boat Lake, which is in line with other studies from supra-permafrost groundwater systems in areas with continuous permafrost (Walvoord and Kurylyk, 2016). The results of our particle tracking simulation also align with studies of water transit times in permafrost areas using tracer methods, indicating that it is common that most of the water that moves through permafrost landscapes is young (e.g. Cochand et al., 2020; Tetzlaff et al., 2018; Throckmorton et al., 2016). However, a high proportion of young water is common in streams in all climate zones (Jasechko et al., 2016), but in continuous permafrost landscapes water ages are even lower than in areas without permafrost (Hiyama et al., 2013; Wang et al., 2022). For example, a similar particle tracking simulation made for a boreal forested catchment without permafrost indicated that most of the water also was between 0.8- and 3.7-years old (Jutebring Sterte et al., 2021b). However, unlike for the Two-Boat Lake catchment, where the maximum age of water was a few years, a considerable fraction (~25%) of the water in the boreal system had an age of between 10 and 1000 years. Older groundwater can be present also in areas with continuous permafrost, e.g., because of a connection to the groundwater system below the permafrost through taliks (Koch et al., 2024). That this effect cannot be seen in Two-Boat Lake is likely related to the recharging nature of the talik under Two-Boat Lake (Johansson et al. 2015a). It should be noted that our model only simulates the age of groundwater in the saturated zone, and that during the snowmelt period runoff occurs also as overland flow in the temporary stream network. Therefore, our simulated water ages cannot be directly compared to results from tracer-based methods.

Even if the groundwater in Two-Boat Lake generally is young, there is a clear seasonal trend with increasing water age as the unfrozen season progresses. This pattern can be seen in the 0-25 cm layer, but it is especially pronounced in the 25-50 cm layer. At the deepest layer, i.e., 50-75 cm, there is no clear seasonal trend, but most of the water that moves in this layer is older than two years. These patterns imply several things. First, the movement of water in the uppermost layer (i.e., 0-25 cm) is mostly restricted to the snowmelt period, and as the thaw depth increases, the groundwater takes deeper flow paths. Similar patterns have been documented from other permafrost regions (Juhls et al., 2020; Lebedeva, 2019; Zastruzny et al., 2024). Second, the turnover time of the water in the uppermost layers is relatively short, but the time that the water has spent in contact with the soil particles increases as the unfrozen season progresses. Third, for the deepest analysed layer (i.e., 50-75 cm) the turnover time is longer, and most of the water has spent several seasons in the saturated zone, i.e., this water has undergone at least one freeze-thaw cycle.

**4.2 The effect of fast and shallow flow paths on the water chemistry**

Most of the variability in water chemistry can be explained by either the amount of weathering products (PC-1) that have accumulated through interactions with soil particles or the degree of evaporative loss the water has experienced (PC-2). Based on the correlations between the PC-1 and PC-2 scores for the surface-water samples and the hydrological variables from the hydrological modelling it seems as if these two components are controlled by separate hydrological factors. For surface water, PC-2 scores – which represent elements supplied with precipitation and that are concentrated through evapotranspiration – show a negative correlation with total runoff.

In spring, when large volumes of snowmelt water move through the system, the chemical signature is more diluted and closer to that of precipitation. The PC-2 scores then progressively increase over the unfrozen season, and the chemical signature of the surface water shifts towards a signature that is more characteristic of the lake water (Fig. 5). That is, as the water from precipitation moves through the catchment, it becomes more concentrated in elements that were present in the precipitation when it fell (e.g., chlorine, Cl). This interpretation is also consistent with the PC-2 scores being positively correlated with DOY, and with water ages increasing as the unfrozen season progresses (Fig. 3). That evapotranspiration is an important process in this dry landscape can be seen in the hydrological model, where two-thirds to four-fifths of the annual precipitation leaves the terrestrial system as sublimation or evapotranspiration. This loss of water translates to a concentration factor of 3-5, which is consistent with the 3.3 times increase in Cl concentration between precipitation and surface water (SI-Table 1). Chlorine is almost exclusively supplied via precipitation and is often used as a conservative tracer for the input of solutes via precipitation (Johnson et al., 2000; Lockwood et al., 1995).

For elements supplied through weathering it is not primarily the amount of runoff that is of importance. Instead, it is the proportion of runoff classified as deep groundwater (i.e., 50-75 cm depth in our model) that has an effect. In July and August, the majority of this deep groundwater had spent more than two years in the saturated zone (Fig. 3). That older and deeper groundwater is richer in weathering products is consistent with studies in other systems and other regions (Jutebring Sterte et al., 2021b; Williams et al., 2015). When looking at the concentrations in calcium (Ca, supplied both via precipitation and weathering), lanthanum (La; supplied primarily via weathering) and Cl in precipitation and surface water they show different enrichment patterns (Deer et al., 1992; Johnson et al., 2000). As mentioned above, Cl increases about 3.3 times between precipitation and surface water and the enrichment can therefore be explained by evapotranspiration alone. For La and Ca the difference is higher, 7.8 and 10 times, respectively (SI-Table 1), which indicates that these elements must also be supplied by an internal process in the catchment soils (i.e., weathering). This interpretation is also in line with the mass-balance budget that Rydberg et al. (2023) developed for Two-Boat Lake, and suggests that La is exclusively supplied by weathering, Cl almost exclusively via precipitation and Ca is supplied by both processes.

Even if DOC was not measured on all sampling occasions, and could not be included in the PCA, the correlation between PC-1, PC-2 and DOC helped us to assess if organic carbon behaves like weathering products or elements supplied with precipitation. For the surface-water samples there is no correlation between PC-1, PC-2 and DOC, but similarly to PC-2, DOC is negatively correlated with the total runoff. The different behaviours of weathering products (PC-1) and DOC could be related to where in the soil profile these products originate. DOC production is highest in uppermost soil layers, where the decomposition of relatively fresh OM is highest (Clark et al., 2008). These surficial layers thaw relatively rapidly, and hence, the DOC source is activated already during the snowmelt period. This means that, like the elements represented by PC-2, DOC is diluted when the runoff is high. However, unlike the elements that are constantly resupplied with the precipitation, DOC in the upper soil layers can become depleted with time, and PC-2 and DOC are therefore not correlated (Stewart et al., 2022). Weathering products on the other hand are primarily produced deeper in the soil profile (Cai et al., 2008; Fouché et al., 2021), and it is not until later in the season when these deeper layers thaw – and the proportion of deep groundwater in the runoff increases – that this source becomes activated. A similar shift in the chemical composition when the thaw in the active layer reaches below the upper organic-rich soil layer, has been observed in other areas with continuous permafrost (Cai et al., 2008; Chiasson-Poirier et al., 2020).

Looking at the soil water sampled using lysimeters and in groundwater wells, PC-1, PC-2 and DOC are all correlated, which could indicate that older, and deeper, groundwater is enriched in both weathering products, elements supplied with precipitation and DOC. One explanation for this is that a high DOC concentration can also lead to a higher solubility of many elements, particularly weathering products that otherwise can have a low solubility in water (Broder and Biester, 2015; Lidman et al., 2017). That groundwater is enriched in elements is also consistent with previous studies that have shown that deeper and older groundwater often has elevated concentrations for a large selection of elements (Clark et al., 2008; Fouché et al., 2021; Stewart et al., 2022). Unlike previous studies in Greenland (Jessen et al., 2014), neither PC-1, PC-2 or DOC showed any correlation with sampling depth, which likely has several reasons. First, the depths where the lysimeters are installed (15 and 30/35 cm) correspond to the two upper layers used in the particle tracking simulation (i.e., 0-25 and 25-50 cm). In these two layers the age of the water is relatively similar, and it is possible that a clearer pattern would have emerged if the deepest lysimeter had been placed in the deepest layer used in the particle tracking simulation (i.e., 50-75 cm) where the water is considerably older. Second, the deeper lysimeters could only be sampled late in the season when the total runoff was low because they are frozen during the entire snowmelt period. The lack of trend with depth could therefore also be a result of this bias in the data. Third, the generally similar age for the two upper layers in the particle tracking simulation suggests that there is a considerable vertical movement of water in the thin unfrozen layer that results in mixing of the water. This mixing would prevent the development of any consistent vertical trend in the chemical composition

## 4.3 Variability in thaw rates during snowmelt on different timescales

Similar to the original modelling period by Johansson et al. (2015a), the runoff is dominated by groundwater, either directly to the lake or via temporary surface streams, but the extended modelling period reveals more between-year variability. Even though the annual precipitation is relatively similar between 2014, 2015, 2016 and 2018 (247, 284, 286 and 261 mm $yr^{-1}$, respectively) the amount of overland flow varies considerably. 2014 and 2015 have a very low contribution of overland flow (less than 15% of total runoff), whereas in 2016 and 2018 around 50% of the annual runoff occurred as overland flow either directly to the lake or via temporary surface streams. These differences in the partitioning between groundwater and overland flow from one year to the other can likely be linked to both the amount of accumulated snow in the catchment and the thaw rate in the active layer, especially early in the thaw season. The amount of winter precipitation (October to March), which dominantly fell as snow was relatively close to the average during the winters preceding the snowmelt periods of 2014 and 2015 (105 and 155 mm, respectively) and the snowmelt started relatively late (mid to late May). This resulted in a low proportion of the runoff leaving as overland flow during these years. In 2016, considerably more snow fell during the preceding winter (220 mm) and the snowmelt started in early April, which resulted in a higher proportion of the runoff leaving the catchment as overland flow. For 2018, the amount of accumulated snow was slightly lower (91 mm) and the snowmelt period started in mid-May, i.e., similarly to 2014 and 2015. Still, during 2018 a much larger percentage of the water left the terrestrial catchment as overland flow (52%) compared to during 2014 and 2015 (7 and 13%, respectively). This can most likely be attributed to the preceding year, 2017, being considerably wetter compared to other years (407 mm $yr^{-1}$ of annual precipitation). This resulted in a high soil moisture and high groundwater levels at the time when the active layer froze in the autumn of 2017. A high

content of ground ice in the active layer implies that the capacity for infiltration during the snowmelt period was
limited, and the latent heat content in the ice likely contributed to a slower thaw rate in the active layer this year
(Clayton et al., 2021). As the ground surface thawed, the high soil moisture content may also have contributed to
saturation-excess overland flow during the late snowmelt and summer periods. For all other years, overland flow
did not occur during summer, presumably because the generally dry conditions (evapotranspiration exceeds
precipitation) normally result in high infiltration. For example, observations in the field confirm that temporary
surface streams only appear during limited periods during the snowmelt period and during wet periods during fall.
While the long-term hydrological modelling indicated that interannual variability in thaw in the active layer exerts
a strong control on runoff patterns, our field observations during the intense monitoring period revealed that – on
a finer spatial scale – the hydrology also controls thaw rates. The thaw-depth monitoring during May 2017 showed
that wet locations thawed significantly faster than drier locations. This would support earlier findings from areas
with continuous permafrost, where variability in the active layer thickness has been attributed to the development
of a hillslope groundwater drainage system (Chiasson-Poirier et al., 2020). The fastest thaw rates and largest thaw
depths in the Two-Boat Lake catchment were measured in the wet locations of the slope and directly under the
temporary surface streams. This would suggest that advective heat transport with surface and near-surface water
plays an important role in determining the thaw rate, as has been observed elsewhere in the Arctic (Dagenais et
al., 2020; De Grandpré et al., 2012; Sjöberg et al., 2016). When compared to earlier investigations in the Two-
Boat Lake catchment, it is noticeable that the maximum measured thaw depth during the intense monitoring period
(48 cm ±9 cm), was almost the same as the average active layer thickness observed under similar vegetation cover
(i.e., wetland) in August 2011 by Petrone et al. (2016). This indicates that the ground in connection to temporary
surface streams thaws rapidly during the snowmelt period, but that the rate of thaw is much slower during the
remainder of the summer season. At the drier location where the ground temperatures are monitored down to 2-
m depth the thaw developed slower. In 2017 the 50-cm sensor reported above zero temperatures on June 9[th] and
the 75-cm sensor on July 27[th]. Taken together with the drier locations having deeper maximum thaw depths
compared to wetlands (70-80 cm; Petrone et al., 2016), this indicates that even if wetter locations thaw more
rapidly in spring due to advective heat transfer, the effect of the latent heat in wet soils becomes more important
in determining the maximum thaw depth later in the season (Clayton et al., 2021).
This fine-scale variability in thaw rates in the active layer was not included in our hydrological modelling and
subsequent particle tracking simulation. The finding that the thaw rate is faster under the temporary stream
network suggests that water that reaches the lake during the snowmelt may interact more with soil particles than
what our modelling indicates. Our simulated groundwater ages may also underestimate the content of older water
earlier in the season, as considerably deeper soil layers are thawed earlier in the season and possibly in a connected
drainage pattern, similar to that observed by Chiasson-Poirier (2020) in northern Canada.
**4.4 The influence of melting ground ice on the chemical composition during the snowmelt period**
Looking at the isotopic composition of the water samples from Two-Boat Lake they were roughly 5 ‰ heavier
than surface waters from Pituffik in northern Greenland (Akers et al., 2024), the Scotty Creek drainage system in
Canada (Hayashi et al., 2004), and surface waters in the Yukon region of Canada and Alaska (Lachniet et al.,
2016); SI-Fig.1). This heavier signal is likely related to dominant wind patterns and relatively warm sea surface
temperatures in the source area for the water (Sodemann et al., 2008). For the stable isotopic composition in

stream samples collected during the intense monitoring period there is a temporal trend that roughly follows the evaporation line (Fig. 5B). This could suggest fractionation during evapotranspiration, sublimation, condensation, or freezing–thawing processes, but because the deviation away from the evaporation line at the end of the monitoring period is directed toward the composition of rain and ground ice it could also be related to a shift in the source of the stream water (Fig. 5B). An influence of melting ground ice would be consistent with the mixing model (Fig. 5C), which suggests that the isotopic signature of the stream water is heavily influenced by melting ground ice.

Ground ice filling the pore spaces in the active layer most likely formed from the water present in the active layer during the previous fall when the ground froze, although some infiltration and migration of meltwater during warm periods during the winter cannot be completely excluded. According to the particle tracking simulation, most water present in the active layer during the fall freeze-up is at least one year old and a substantial fraction of the water is older than four years. In light of this, the stable isotopic composition of the ground ice falling somewhere in the middle between those of snow and rain is to be expected and has also been found using similar methods in the continuous permafrost zone in northern Canada (Tetzlaff et al., 2018, Wilcox et al., 2022). In comparison to findings in lowland polygonal tundra in the continuous permafrost zone in Alaska, where winter precipitation did not contribute to the stable isotopic signature of active layer pore waters, water in the active layer in Two-Boat Lake appears to be more well mixed at least in the fall (Throckmorton et al., 2016).

Similar to the stable isotopic signature, there is a shift in the chemical signature during this period. PC-1 increased, PC-2 first increased and then remained stable, and DOC first decreased and then increased slightly. The increase in PC-1 is consistent with patterns observed over the entire unfrozen period, and it increases as the proportion of deep groundwater increases. For PC-2 and DOC, the trends during the intense monitoring period do not show the same negative correlation with total runoff as during the entire unfrozen season. For PC-2 the initial increase from May 19th to May 25th, when the total runoff decreases, is consistent with the expected trend (Fig. 5A). However, when the total runoff then increases again to May 31st, the PC-2 scores remain virtually the same as on May 25th. The DOC decreases when the total runoff decreases from May 19th to May 25th, and it then increases slightly until May 31st. Even if the PC-2 scores for the last two sampling occasions are similar, the long-term data would suggest a stronger response to changes in the total runoff. However, the chemical and stable isotopic signature is not merely a question of more or less water, we also need to consider where the water comes from. From the stable isotopic signature of the ground ice, we can see that there is a considerable influence of melting ground ice over the intense monitoring period. The water from the melting ground ice will not only be isotopically heavier ($\delta D$ and $\delta^{18}O$) compared to the meltwater from snow, the pre-event water that has resided in the ground for an extended time will also be enriched in elements related to both PC-1, PC-2 and has higher DOC concentrations.

That ground ice and the thaw of permafrost has a profound effect on the chemical signature of soil water in permafrost regions has also been reported from sites in Alaska (Fouché et al., 2021), and that pre-event water and elements that can be flushed from surficial soil layers have been shown to be important also under a wide variety of environmental settings (Fischer et al., 2017; Juhls et al., 2020; Ross et al., 2017). Furthermore, presumably most of the melted ground ice comes from the near stream zone, where the thaw rate is highest. This suggests that these temporary "riparian" zones are important for the evolution of the stable isotopic and chemical signature of the surface water, which is analogous to the importance of the riparian zone in other systems (Jutebring Sterte et al., 2022; Lidman et al., 2017).

**Conclusions**

Our monitoring of thaw rates in the active layer during the 2017 snowmelt period shows that, even if drier areas tend to have a thicker active layer by August (Petrone et al. 2016), areas in connection with temporary surface streams thaw more rapidly in the early stages. The rapid formation of an unfrozen zone beneath these temporary streams allows meltwater to interact with soil particles and pre-event water already early in the snowmelt period. This is reflected in, e.g., the stable isotopic signature of stream water, which initially resembles that of snow but quickly shifts toward a signature more similar to melting ground ice.

Based on the hydrological modelling, we can also conclude that although interannual differences in the hydrological conditions can result in up to half of the runoff occurring as overland flow in some years, runoff to Two-Boat Lake is generally dominated by groundwater. The particle tracking simulation shows that while most of this groundwater is young–i.e., less than one year–indicating limited interaction with soil particles, a considerable fraction of the water in deeper soil layers (i.e., 50-75 cm) is older than three years. This "deep" groundwater plays an important role for the chemical signature in the runoff to Two-Boat Lake, especially for elements released through weathering.

However, although the interaction between the runoff and soil particles is important, it is not the only process influencing the chemical signature of the runoff. For elements supplied via precipitation, our data suggest that extensive evapotranspiration in this dry landscape also strongly affects the concentrations in the runoff. Taken together, this indicates that even if the runoff is dominated by the snowmelt period, the groundwater generally is young, and the continuous permafrost restricts water movement to the thin active layer, there remains a strong connection between terrestrial processes and the chemical signature of the runoff.

This connection is especially important when considering the substantial variability in both hydrological conditions and thaw rates—both spatially and temporally—and it highlights the importance of accounting for the effect of soil processes and mixing with pre-event water when assessing water quality and element transport in Arctic landscapes with continuous permafrost.

**Data availability**

The data used in this study is available mainly through three publications, Johansson et al. (2015b), Lindborg et al. (2016) and Petrone et al. (2016), each with an adjoined Pangaea database (doi:10.1594/PANGAEA.845258, doi:10.1594/PANGAEA.860961 and doi.pangaea.de/10.1594/PANGAEA.836178, respectively). For Johansson et al. (2015b) and Lindborg et al. (2016) additional data has been made public through additional Pangaea databases that are linked to the original publications. Meteorological data from the automated weather station (labelled KAN_B) is available via www.promice.org.

**Author contribution**

**JR**: Conceptualization, Formal analysis, Investigation, Writing – Original draft, **EL**: Conceptualization, Methodology, Formal analysis, Writing – Review & Editing, **CB**: Formal analysis, Writing – Review & Editing,

**BMCF**: Writing – Review & Editing, **TL**: Conceptualization, Investigation, Writing – Review & Editing, **YS**:
Conceptualization, Formal analysis, Investigation, Writing – Original draft.
**Competing interests**
Ylva Sjöberg is a member of the editorial board of The Cryosphere
**Acknowledgement**
This study was funded by Swedish nuclear fuel and waste management company (SKB) within the Greenland
Analogue Surface Project (GRASP) and CatchNet project. We also thank two anonymous reviewers for valuable
input during the review process.

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
