# Peer review of "The coupling between hydrology, the development of the active"

_EGUsphere, 2024_

## Author Response (AR1)

**Response to comments from reviewer 1**

**Review of the article "The coupling between hydrology, the development of the active layer and the chemical signature of surface water in a periglacial catchment in West Greenland"**

The manuscript of Rydberg and colleagues presents actual results in studies of processes in permafrost hydrology, groundwater, chemical signature, and isotopes. The greatest advantage of the manuscript is presenting results of own investigations of periglacial lake's catchment in a melting period. The presented field measurements and analytical results provide some important aspects and testify to the authors' knowledge of the hydrochemistry and modeling.

General conclusion. The manuscript is recommended for publication after major revision according to following comments and notes. I believe the manuscript to be an interesting and good contribution. The manuscript will be useful for international Polar community on hydrology and complementary sciences studies.

We thank Reviewer #1 for the constructive feedback and are happy the reviewer appreciated our manuscript. Please find below our responses to the individual comments and suggestions (reviewer #1 comments in *blue* font, with our, author response (AR) in black font).

1. RC structure:

The authors apply quite a number of the different methods that could be better structured in the paper. Interpretation of the results is not clear enough due to complexity of overview, different timescales, or/and visualization.

**AR:** We have address this issue in the revised version. For example, by separating the measured data from modelled data (as suggested by reviewer 1 below) by using the following structure in the method section.

2.1 - Study area

2.2 Modelling

       2.2.1 Hydrological modelling

       2.2.2 Particle tracking simulation

2.3 Measurements and sample collection

       2.3.1 Monitoring of thaw in the active layer

       2.3.2 Meteorology and ground temperatures

       2.3.3 Estimate of ground ice

       2.3.4 Samples for chemical analysis

2.4 Data analysis

       2.4.1 PCA analysis

       2.4.2 Correlation analysis

As for the presentation and visualization of the results we will follow the suggestions of reviewer 1 comments regarding the figures (see further below).

As for the discussion we have kept the general structure – partly because we feel that it follows a logic progression, and partly because reviewer 2 does not seem to have the same issues with the structure (but with the language). However, we have added additional headings to help the reader navigate through the discussion.

4.1 Young water and shallow flow paths dominates the runoff

4.2 The effect of fast and shallow flow paths on the water chemistry

4.3 Variability in thaw rates during snowmelt on different timescales

4.4 The influence of melting ground ice on the chemical composition during the snowmelt period

In addition, we have also updated the text to comply with the specific issues by the two reviewers.

2. RC discussion:

This paper provides new insights into periglacial surface and under-ground hydrological processes of Western Greenland, however, in chapter "Discussion" there is no enough discussion of some important questions in Polar hydrology in larger scale since these issues are important for the whole Arctic and not exclusively for Greenland.

**AR:** We have expanded the discussion to also include comparisons to findings from areas outside Greenland, based on the reviewers' input. We have also, following comments from both reviewers, clarified and expanded the discussion. Also see the responses to the specific comments below.

3. RC objectives:

In "Objectives" authors hypothesizing about influence of freeze-thaw processes and hydrology (including water-age) to chemical and isotopic signature of surface water and groundwater in the lake catchment in West Greenland. This question is not new in permafrost hydrology. That is why then hypothesizing you should be precise with own study novelty justification. Better focus on own field measurements. There is not much information available in the literature concerning detail measurements of active layer hydrodynamic and chemistry during melting period. This is can be the core to build the paper around and shift the focus on analysis of the obtained data rather than modeling.

**AR:** We agree with the reviewer that the main novelty is in the direct observations of the active layer hydrodynamics and chemistry during the melt period, but we also argue that the combination of detailed field observations with a robust modelling framework for interpreting the mechanisms behind the observed dynamics in hydrology and chemistry is a novelty. In the revised manuscript we have tried to clarify this further in the aim of the paper, to highlight the specific contributions of the study to current state of the art.

4. RC reorganization:

The manuscript needs to be partly re-organized. Obtained data and modeling could be presented separately. "Method" paragraph should be better structured as well as "Results". Sometimes authors combine the data obtained by different methods, which significantly complicates the perception of the results. In my mind paper will benefit from separating the results of measurements and model calculations, and later you can discuss altogether in the "Discussion" chapter.

**AR:** In the revised manuscript, we have followed the reviewer's suggestion to more clearly separate the modelling and empirical data in the method section, to ensure a clearer structure for the reader (see above).

5. RC Figure 1:

needs significant revision as it lacks much of what is mentioned in the text. The surface temporary streams should be shown on the map. Isolines of topography could be added as well for easier water flow direction understanding. The legend on the figure itself and the figure caption should be unified: "automatic weather station (AWS)" or "weather station"? "Ground temp" in the legend and "GW-12" in the texts has the same meaning? Is the point of "ground temperature" measurements and points of "lysimeter" and "groundwater well's location" the same? Where is "AL-transects" on the map and in the legend? The lake name "Two-Boat Lake" should be added on the part B (there is no label "B" on the figure as well) and it should not be "TBL" as it is on the part "A".

**AR:** We find that all these comments are fair and constructive for improving Figure 1. In the revised manuscript, we have corrected Figure 1 according to the reviewer's suggestions. We have also changed the background map to a regolith map instead of the vegetation map.

6. RC Figure 3 and 4

Figures numbers need to follow the order as it is in the text. Figures 3 and 4 should be rearranged to match the text.

**AR:** Thank you for pointing this out. To maintain consistency, we have fixed this.

RC Vegetation

What type of vegetation zone is in the TBL catchment? Short explanation can be added to the paragraph 2.1. The type of vegetation is very important for evapotranspiration calculation and modelling as well as DOC variation analyse.

AR: This information is included in the revised version of the manuscript.

7.    RC Lysimeter

Explanation of lysimeter measurements should be provided in the text. Methods of soil/ground water sampling should be added in the text with more detail explanation: number of samples, time and depth of sampling, methods and instruments of water sampling; what kind of filters were used; how the samples were preserved and transported etc.

**AR:** We have Added more details to the description, and we have tried to make the link to Lindborg et al., (2016) more visible.

8.    RC walking

Paragraph 2.2. During field work, repeated walking to take measurements and samples can by itself significantly affect the depth of thawing and even the chemical composition of the water. Was it taken into account in any way, either during field work or later in the data analysis?

**AR:** For the thaw depth measurements during the melt period of 2017, we walked along the measured transects seven times in total and took care to not walk on the exact line that we measured. While we could not notice in the field, any impact on the ground from where we walked, we cannot completely exclude any impact from our walking on our observations, however, we believe it to be very small if at all there. The valley where we work is regularly (we believe several times daily) traversed by reindeer and muskox and only at a few locations higher up in the terrain, are tracks visible. We therefore think it is unlikely that our walking during the sampling would have significantly impacted the thaw depth, beyond what is normal from wildlife activity in the area, and we do not believe it would impact the observed trends in thaw depth over the observation period. No, correction for such impact was made in the analysis. We have added a brief comment on this in the revised manuscript.

9.    RC TDR

The depth of TDR sensor location should be added into the text. The number of Ground temperature (GT) points are not clear on figure 1: there are 2 rings on a sector A and 3 rings (blue and black) on a sector C. Are all rings GT and TDR loggers? Could you add the information on a figure and to the legend? There is information about four locations on sub-catchment for TDR sensors (line 294), but no one of it is in the legend of Figure 1. This needs to be adjusted.

**AR:** We have added the requested information to Figure 1 We have also added more details regarding this and tried to make the link to Johansson et al., (2015b) more evident.

10.    RC Ground ice

The purpose of paragraph 2.4 of the Methods section (ground ice) is unclear as it is not subsequently presented in either the Results or the Discussion.

**AR:** The results of these calculations are described in paragraph 3.3 and provides further evidence of the magnitude of contribution of water from melting active layer ground ice to flow in the active layer during the melt season. We have revised the text here to clarify this better.

11.    RC  Mike

The model MIKE SHE and its applicability for permafrost zone should be better explained. For example, Line 161-162: "…four layers in the underlying permafrost (1-200 m depth)…". Is this a typo? Could it be - 200 cm? If this is indeed the case (200 m), then modeling permafrost to such depths requires a more detailed explanation. In that case, how it is used for surface runoff and groundwater (top 100 cm) modeling? Are the differences between thawing of permafrost and melting of ground ice? Did authors (or Johansson) validated the model for permafrost condition and specific hydrophysical processes? In my opinion, it should be highlighted in the methods. Referring to Johansson et al. (2015b) is not really enough since it is quite important for this particular study.

**AR:** We have added more information regarding this, however, to explain the entire set up is not really feasible considering that there are several complete papers dedicated to just describing how MIKE SHE was adapted to handling permafrost (Bosson et al. 2012, 2013), the collection of data in the Two-Boat Lake catchment (Johansson et al., 2015a), and the design and validation of the hydrological model for Two-Boat Lake (Johansson et al. 2015b). We have tried to make sure that these references are more visible and provide more detail regarding how the model works

12.    RC Mike SHE and groundwater age

Estimation of groundwater age is not fully clear due to used model type (Paragraph 2.6). MIKE SHE does not verified for permafrost conditions as it is mentioned in the text. The model has been tested for the boreal zone, which may be the reason for an inaccurate estimate of the groundwater age for the permafrost zone. Stable water isotopes content could help in a groundwater age estimation. The stage of hydrological cycle and, as a consequence, groundwater age directly influences the isotopic composition. Was the obtained data on isotopic composition analyzed in this regard? The age of groundwater was simulated for period 2016-2019 (Paragraph 3.2), that includes May 2017 when samples for isotope analysis were collected.

The number of collected samples for water stable isotopic composition, unfortunately, is not really sufficient for statistical analysis. In this regard, it would be even more important to conduct a comparative analysis with known data on other Arctic regions.

**AR:**    Here, we disagree with some of the comments from the reviewer, and the reason for the disagreement is likely that we have not explained our model procedure well enough. In fact, we would argue that the MIKE SHE-model that we use here is one of the best validated models for catchment-scale hydrology in a continuous permafrost system. While the MIKE SHE-model in itself does not simulate freeze-thaw dynamics, it uses observed (prescribed) ground temperature data to dynamically adjust the hydraulic properties of the ground to mimic the effect of permafrost and seasonal freezing on the hydrology. This modelling approach has been successfully calibrated and validated for the Two-Boat Lake catchment (e.g., against, e.g., changes in lake level as described in Johansson et al. 2015a). This parameterization of permafrost is crucial to our objectives, which are not to investigate the dynamics of permafrost itself, but the impact of such dynamics (as observed and prescribed) on the hydrologic system. We understand that we did not make this clear enough in the original manuscript and will provide a better motivation for the choice of model strategy in the revised version.

The age of groundwater in the active layer is modelled through the particle tracking function in MIKE SHE. Our trust in the groundwater age estimates from the particle tracking model builds on the successful validation of the hydrologic flow model (Johansson et al., 2015a) that provides the basis for this model calculation. Even though we trust the overall results from the particle tracking, these age estimates are of course associated with some uncertainty especially when comparing outputs with samples collected at point locations and at specific dates that may not correspond to exact times and locations in the model.

We do not have the long time series of water isotopic data needed to compare age estimates of water in the active layer with the modelled water ages (as was done with a very similar model setup in Jutebring et al., 2021). We are not sure how the reviewer means that we could have calculated transit times of water from our limited sampling of water stable isotopes in the spring of 2017. However, it is clear that these data indicate that a substantial part of the flowing surface water originate not from the preceding winter's snow but must be at least from the summer before. But these samples are also taken from overland flow water, so it is difficult to make more than anecdotal comparisons to simulated groundwater ages (such as in the end of section 4.2).

Regarding comparing the particle tracking results to the isotopic data, it is a bit tricky. The particle tracking uses the hydrological conditions for 2016-2019, but it cycles this period for 100 years. Hence, it is questionable if the conditions at the end of the modelling simulation is comparable to samples collected during a single year (i.e., 2017). Furthermore, we analysed the water samples solely based on their isotopic composition. To determine the transit time of the water, we would have needed to analyse the tritium composition or conduct precipitation or stream sampling at weekly, bi-weekly, or monthly intervals. Although this additional information would have been useful, we could not perform these analyses due to logistics, and therefore we cannot make definitive statements regarding transit time. While the available data are limited in space and time, comparing the isotopic compositions from Two-Boat Lake with values reported in the literature offers a useful way to assess the relative positioning of the samples. However, this comparison will not provide additional information on the transit time. We have added an SI-figure that includes both more data from our study and other studies (see further below).

13.    RC Figure 5B

Figure 5 (B) shows the graph of isotopic composition with only one value (point on the graph) for the rain and only one for the snow. Nevertheless, precipitation samples were collected in 2011, 2014, 2017, and 2019 as it is noticed in Paragraph 3.5. Are all the samples having the same value? Is it a mean value, may be? What about ground ice sample (Line 316) – what was the sampling depth? Line 317: "...winter (snow) and summer (rain) values...". Why the rain strictly attributed to summer? Liquid precipitation occurring in spring (as in May 2017) and autumn and may have different isotopic composition.

Since DOC discussed in the text and provide an important information about water geochemical composition, more detail explanation is required (number of samples and sampling approach, filtration and conservation).

**AR:** The value in the graph is the mean value that was used for the mixing model. The error bars represent the variability between samples (not measurement errors). This will be clarified in the revised version in the figure caption.

We have also provide a bit more detail regarding the DOC samples in the revised version

14.    RC Figure 5A

The figure 5 (A) with PCA analyze is fully overloaded. Lack of direct access to data does not allow reader to evaluate the reliability of the results. I would recommend to add the table with geochemical (and DOC) data as supplementary material or you may think about to place it to the open database, for example, Pangaea as used before (Johansson, Lindborg, Petrone et al.).

**AR:** Except for the samples from 2017-2019 (which are provided in the SI-material for review purposes) all data that is asked for is available via the Pangaea data base that is attached to Lindborg et al. 2016. The intention is (as stated in the cover letter) to make also the remaining data available through a Pangaea database (and provide a link from the original Lindborg et al. 2016 database to the additional one).

Regarding the figure being "overloaded" we are not sure how to solve this problem, and hence the figure has not been altered. We will figure caption to better explain both the rationale behind the figure and the interpretations that can be made using it (see also comments to reviewer 2).

15.  RC Discussion

The main issue of a "Discussion" is water-age in a top active layer. The 1-year age is not surprising if you are sampling in May when snowmelt water partly flows down the surface and partly infiltrates to a ground. Meltwater is almost the only source for the runoff formation in May. The 3-4-years age of water is more interesting but the modeling results could be supported by stable water isotope values. Unfortunately, it could not be considered in the paper, possible due to lack of data. Relation of DOC and water age as well as vegetation and soil types are also not a novel just by itself. It seems to me necessary to elaborate on the formulation of the main conclusions, focusing on the novelty of this particular study.

**AR:** As indicated and stated before, constrained by logistic the water samples were collected only during a short period by, which it was not possible to determine the transit time of water from the water isotopic composition.

Special Comments:

16.  RC figure 1.1:

The legend to Figure 1 includes "Vegetation types" but "Water", "Grassland", "Barren", and "Wetland" are not exactly vegetation types. Several vegetation complexes/microlandscapes could be on a wetland. The legend is recommended to be restructured better.

**AR:** The vegetation types used comes from the classification used in Clarhäll 2011 (and throughout the GRASP project), however, we agree they are not established and well-defined vegetation types. Because the vegetation types are not central to the study we have replaced the vegetation map with a regolith map (which also indicate where wetter areas are located).

17.  RC figure 2:

The period of intense measurements could be added on the figure 2 for understanding of source of data - field or modeled.

**AR:** This have been added to the revised version.

18.  RC figure 5:

The x-axis should be added on figure 5 (C).

**AR:** Not fully clear what the reviewer asks for here. All panels have x-axis (with axis lables and titles)

19.  RC figure 5 lables:

The y-axis on figure 5 labeled as "Fraction of runoff from ground ice or rain" but it is different in the figure caption - "fraction of melting ground ice and rain..." The label of the y-axis should be unified.

AR: This have been fixed in the revised version.

20.  RC figure 5 (c):

The figure 5 (C) is not really informative. Would be enough to discuss it in the text? The increase of groundwater fraction content of 0.2 in 12 days fits within the limits of measurement error.

**AR:** In the end we chose to retain this part of the figure

**Response to comments from reviewer 2**

**Review of "The coupling between hydrology, the development of the active layer and the chemical signature of surface water in a periglacial catchment in West Greenland" for The Cryosphere.**

**General comments:**

This is a well-orchestrated study of hydroloigic sources and flows into a small lake in southwest Greenland. The article would be of interest to permafrost scientists, hydrologists, and geochemists. The study design appears well done and is presented in a clear fashion.

However, as written this paper needs a lot of work- major revisions. There to many typographical errors, strange wording, and odd punctuation. It is confusing at times and it is not fair to ask a Reviewer to make the types of edits needed to make this more readable. I also have some issues with specific misuse of terminology (like "active layer") and the statements ascribing evaporation as a main source of increased lake water ionic strength. This paper can probably be fixed but it will require some concerted efforts, more reading of similar literature, and a reframing of the results and conclusions.

**AR**: We thank Reviewer #2 for the constructive feedback to the manuscript. We corrected the typographical errors, clarified wording and improved punctuation. Based on our interpretation of the reviewer's comments, we have addressed what we understand to be the key points raised. We hope that our responses sufficiently clarify and resolve the concerns of Reviewer #2. Please find below our responses to the individual comments and suggestions (reviewer #2 comments in *blue* font, with our, author response (AR) in black font).

**Comments keyed to the text:**

1. RC2 17:

   surface water

   **AR:** have be changed

2. RC2 17:

   What is meant by "old groundwater"? Perhaps just say "ground water and shallow subsurface flow" here in the first sentence. A few lines later you say it is 4 years old so it is not old.

   **AR:** We agree and changed to groundwater contribution in the revised version

3. RC2 22:

   stable isotopic

   **AR:** agree we changed to stable water isotopic composition

4. RC2 23:

   what are hydrological situations? Events?

   **AR:** "The term 'hydrological situations' is indeed unclear. We have revised this part

5. RC2 28:

   similar to other

   **AR:** have been changed

6. RC2 31:

   Rainfall typically has little in terms of major ionic load. Look at your own data... Soil interactions are the dominant source of ions in waters. Thousands of papers show this. Including some in the Arctic/boreal.

**AR:** We are not fully sure what the reviewer asks for here. Even if the interaction with the soil is the major source for many ions it cannot be disregarded that the precipitation also carries ions (e.g., chlorine and long-range pollution) that will affect the chemical composition of the water. That soil interactions are important is also pointed out in the following sentence, and the idea here is to put the processes in a chronological order as they affect the composition of the water (from the atmosphere and as the water moves through the system).

7.     RC2 32:

through a catchment

**AR:** have been changed.

8.     RC2 33:

and bedrock

**AR:** have been changed.

9.     RC2 37:

projection instead of prediction. We do not "predict" like a Tarot card reader. We project results or model outcomes.

**AR:** We have rewritten this part..

10.     RC2 39:

What is "the atmospheric state"? Conditions?

**AR:** have been changed.

11.     RC2 53:

There are many studies in the boreal showing the source of this DOC is surface dead or dying organic matter and vegetation. For example:

Cai Y, Guo L, Douglas TA. Temporal variations in organic carbon species and fluxes from the Chena River, Alaska. Limnology and Oceanography. 2008 Jul;53(4):1408-19.

But this may be more germane since the study area is perhaps actually "quasi-boreal/tundra" (?):

Holmes RM, McClelland JW, Raymond PA, Frazer BB, Peterson BJ, Stieglitz M. Lability of DOC transported by Alaskan rivers to the Arctic Ocean. Geophysical research letters. 2008 Feb;35(3).

They show an increase in DOC with spring melt and with summer rain events. Also much of the boreal is discontinuous permafrost where many river shave year long flows. It may be better to focus on watersheds similar to the Greenland one that do not have substantial (or any?) groundwater flow in winter/late winter.

**AR:** We thank the reviewer for pointing us to these articles, and we agree that the source of the DOC is dead plant material. We have revised the text and incorporated the Cai et al. reference to this section.

12.     RC2 59:

polygonal

**AR:** have been changed.

13.     RC2 60:

Similarly,

**AR:** have been changed.

14.     RC2 63:

and riparian

**AR:** have been changed.

15. RC2 64:

    permafrost is not thawing in early spring. Do you mean thaw of the seasonally frozen and thawed layer?

    **AR:** True, this has been changed.

16. RC2 65:

    I would put overland flow first as it is far more runoff volume than evaporation

    How about "over the course of the summer thaw season" instead of "in time"?

    **AR:** We have changed the order of overland flow and evaporation, and changed "in time" to "over the course of the thaw season".

17. RC2 82:

    the active layer is relatively thin in some areas but not everywhere. Maybe introduce continuous permafrost/high Arctic/cold here to frame the study catchments? I think the study area is in continuous permafrost yet many of the boreal studies used in the Introduction are discontinuous permafrost related. Does the study area have year-round groundwater flows? If not then, again, the Introduction needs to present studies of similar areas. Perhaps high Arctic Canada or Toolik Lake, Alaska or some of the Russian catchment studies?

    This one:

    Ma Q, Jin H, Yu C, Bense VF. Dissolved organic carbon in permafrost regions: A review. Science China Earth Sciences. 2019 Feb;62(2):349-64.

    **AR:** Thanks for pointing out we have added more detail regarding if the studies we refer to are done in areas with continuous, discontinuous or non-permafrost areas. This will hopefully make the comparison to between studies easier to follow.

18. RC2 91:

    Despite being located

    **AR:** have been changed.

19. RC2 92:

    ) and little spatiotemporal data being available due to the region's

    **AR:** have been changed.

20. RC2 94:

    means that the processes

    **AR:** have been changed.

21. RC2 95:

    covary

    **AR:** have been changed.

22. RC2 120:

    and active layer… were collected

    **AR:** have been changed.

23. RC2 125-125:

    this does not describe what is in "A"

    **AR:** have been changed.

24. RC2 126:

    red outline

    **AR:** have been changed.

25. RC2 127:

I do not see a "B" labeled on the map areas. Use capital letters in the caption to be consistent.

**AR:** A "B" have been added, and we will use capital letters in the figure caption in the revised version.

26. RC2 128: C is to the right? What are the blue and black circles in C?

This Figure caption makes little sense...

**AR: We** have tried to fix this in the revised version.

27. RC2 138:

no comma needed before were

**AR:** have been changed.

28. RC2 139-141:

has this "error estimate" protocol been used by anyone else? If so, it would be good to cite it. Surface elevation heterogeneity is probably a larger factor than anywhere "rising up" between measurements. What is the value of this error measured?

**AR:** have removed this part in the revised version.

29. RC2 145:

provide a summary of the thermistor depths

**AR:** This information will be added in the revised version.

30. RC2 148:

water was released

**AR:** have been changed.

31. RC2 156:

that are active

**AR:** have been changed.

32. RC2 158:

was estimated

**AR:** have been changed.

33. RC2 165:

zone from (no comma)

**AR:** have been changed.

34. RC2 179-180:

reach downstream

**AR:** have been changed.

35. RC2 201:

dissolved organic carbon

**AR:** have been changed.

36. RC2 204:

simultaneously with monitoring

**AR:** have been changed.

37. RC2 207:

and the snowpack

**AR:** have been changed.

38. RC2 208:

stable water isotopic

**AR:** have been changed.

39. RC2 211:

were thawed

**AR:** have been changed.

40. RC2 221:

data was then

**AR:** have been changed.

41. RC2 223:

indicating

**AR:** have been changed.

42. RC2 223:

they were not or they were? Why were they removed if they were not contaminated?

**AR:** We have changed this part to make more sense.

43. RC2 224:

a groundwater well

**AR:** have been changed.

44. RC2 259:

the vertical line?

*Use capital letters in the caption.*

*Do you have precipitation information? Rain events would be particularly good to add to this.*

**AR:** We have changed to vertical line and added precipitation in a separate panel (as rain and snow). Based on reviewer 1's comments we have also indicated the intense study period in the figure.

45. RC2 Figure 4.

This is a really great representation!

**AR:** Thank you and we are happy you appreciated Figure 4.

46. RC2 285:

Equivalent

**AR:** have been changed.

47. RC2 286:

or evaporated? Perhaps as meltwater from snow pooled/moved across the landscape some was lost to evaporation?

**AR:** We estimated the evaporation using the hydrological model, which has been clarified in the revised version.

48.   RC2 294:

no comma needed after sub-catchment

**AR:** have been changed.

49.   RC2 307:

monitoring period

 **AR:** have been changed.

50.   RC2 309:

here and throughout: what you are measuring is not actually the "active layer" it is the depth of seasonal thaw at a given point in time. The true active layer would be the thaw depth in late August/early September. Please go through EVERYWHERE that you talk about the active layer and clean up spots where it is not the actual active layer being measured.

**AR:** Yes, true. We have checked the terminology in the revised version.

51.   RC2 311:

I recommend saying "stable water isotope(s)"

 **AR:** have been changed.

52.   RC2 Figure 5B.

Your stream water samples seem to be exhibiting evaporation from the "meteoric water line" identified as the line linking the snow and rain. This would be the preferential increase in d18O relative to dD in the samples.

Look at the d-excess values. The lower d-excess values in lake water suggest evaporation compared to rain and snow likely occurring between melt and flowing downslope to the lake. But I think over all the use of evaporation to get the lake ionic concentrations to where they are is overblown. Interactions with mineral particles/weathered surfaces is likely a larger source of ions.

Evaporation is not surprising at all but it must be considered as both a source of lost liquid (and thus higher major ion concentrations) as well as a means of altering the "mixing" between snow and rain and their relationship to the stream water and the lysimeter/ground ice. There are many citations you could pull in to address this. I am not sure how it would affect your mixing/fraction of melt calculations would be affected by this but you cannot ignore it.

**AR:**  It is correct that stream water samples deviated from the Global Meteoric Water Line (GMWL). While it is commonly assumed that rainfall has a deuterium excess (d-excess) of around 10‰, studies by Pfahl and Sodemann (2014) and Sodemann et al. (2014) have shown that in Greenland, d-excess values can range between 3‰ and more than 10‰. Similarly, our rainfall observations revealed a wide spread in isotopic values, with also observed d-excess of 5‰. This variability makes it less straightforward to attribute d-excess directly to evaporation processes e.g., in stream samples.

Some snow samples exhibited d-excess values equal to 5‰, suggesting limited evaporation. Conversely, certain samples showed lower 5‰ indicate evaporative influence, likely due to its aspect. Stream water samples showed a distinct deviation from the GMWL, with lower d-excess values between -2‰ and 4‰. Over time, the isotopic composition of stream water shifted from a snow-dominated to a rain-dominated signal. We will include an evaporation line, estimated following Bennetin et al. (2018), to clarify that stream water samples—rather than undergoing further evaporation—tend to evolve toward the isotopic signature of rain and soil water.

In contrast, lake water from the Two-Boat Lake exhibited a very low d-excess of approximately -18‰, indicating substantial evaporative enrichment well beyond what was observed in precipitation, soil, or stream water samples. Values for the lake and previously published data from the region will be incorporated in a new SI-figure (see comment to reviewer 1 above), however, because the focus of this manuscript is on the runoff and processes in the terrestrial part of the catchment we feel that going into too much detail regarding the lake, and lake processes, is beyond the scope of this paper.

53. RC2 348-350:

I am not sure how you are stating the higher ionic concentrations in the lake are from evaporation. Yes, some evaporation is probably occurring in the lake and between melt/tundra and lake but the majority of the surface water likely is picking up ions along the way. Also, the bottom of the lake has a talik (as you say) and clearly the waters in the lake can mix with those mineral soils. As well as wind-driven mixing of soils and weathered materials during the open water season.

Maybe provide a plot(s) of some of the major ion concentrations in your different sample types- not just the PCA. Rainfall and snow typically have little to no ionic load. Attributing the increases in the lake form evaporation is likely not warranted.

For example, in looking at your data: Let's use Ca.

Lake mean values: 2723

Precipitation mean values: 15717 micrograms/L

So that is a 7X enrichment... Is that much evaporation occurring? Probably not.

You could use Cl as a conservative tracer and look at Cl:X ratios of your ions. What does Cl do between precipitation and lake concentrations and is this all from "evaporation"?

Also, the isotope results show some evaporation but this is of the stream water not the lake. Likely the lake has some evaporation, too. Perhaps show the lake waters on the d18O-dD plot to see if there is fractionation that suggests evaporation? based on D-excess of lake showing much evaporated samples

**AR:** Yes, for several of the major ions the interaction with soil particles is certainly a major source, but it is not the only source. We have tried to make sure that the interaction with soil particles is put forward a bit more in the revised version. However, from the mass-balance model for Two-Boat Lake (Rydberg et al. 2023), it is evident that precipitation is a considerable source for, e.g., Ca, Na, Mg, K, and basically the only source for the halogens. The mass-balance budget only looks at the net result of several processes, but it still indicates that precipitation is an important source and that the input with precipitation actually accounts for the major part of input of these elements. Furthermore, the idea that low precipitation and high evaporation is a driver for the concentrations of ions in lakes in the Kangerlussuaq region is rather well-established (see for example Ryves et al. 2002).

From the hydrological model it can also be seen that he ET from the terrestrial parts ranges from138–199 mm per year (precipitation is 163-366 mm per year), and the inflow to the lake is in the range 25–159 mm per year (normalized over the terrestrial catchment area; average used for the mass-balance budget is 73 mm). Hence, during a dry year almost all water leaves the terrestrial part of the catchment as evapotranspiration (a concentration factor of 2.5 to 6.5 for wet and dry years, respectively). This massive loss of water volume naturally affects the concentration of any ion that was present in the water when it was deposited. Furthermore, on average 7 mm of water leaves the lake through the outlet, which means that 90 % of the input during an average year leaves the lake as evaporation. Actually, even though the hydrological model suggests that outflow occurs every third year, we have not observed any surface outflow from the lake during the entire study period (2010-2019). This means that for extended periods of time the lake is to be considered as a closed basin lake (the residence time for a solute in the lake is more than 300 years). However, there is occasional outflow, and the lake is therefore much less saline than many other lakes in the Kangerlussuaq region, which are classified as oligo-saline (Ryves et al. 2002).

We have tried to revise the manuscript to, i) better explain the dry nature of this landscape (both in the site description and when discussing the effect of evapotranspiration), ii) elaborated on the results from the PCA and how the different elements relate to the two PCs and what we then compare to the surface water flow and the "deep" ground water iii) emphasize that the interaction with soil particles is an important feature also in this landscape, iv) we have also added a table to the SI with concentrations for three key elements and how they vary between sample rain and surface water (because the paper focuses on the terrestrial system we do not feel that comparing with the lake values makes sense, because then there are also a number of factors related to the lake that needs to be considered, e.g., precipitation in the water column when OM is degraded, further evaporation and the long residence time of solutes in this lake).

54. RC2 357:

on May 31st

**AR:** have been changed.

55. RC2 380:

control

**AR:** have been changed.

56. RC2 385:

fall within

"both" is not needed (three items follow it)

**AR:** have been changed.

57. RC2 386:

 in runoff

**AR:** have been changed.

58. RC2 408:

 0-25 cm what? Soils? Soil column?

**AR:** Yes, 0-25 cm below the ground surface. We have  clarified this in the revised version.

59. RC2 412:

here and throughout: unless the journal states otherwise references are typically presented in chronologic order. This seems to be reverse last name alphabetical?

**AR:** have been fixed.

60. RC2 420-424:

No. The waters interact with surface soils and vegetation and pick up ions. This is not an evaporative process. You can calculate this: take whatever concentration something is in rain or snow and see how many "X" you would have to evaporate it to get it to the concentrations in the lake. You are totally ignoring soils and vegetation along the flow to the lake and with residence time in the lake as potential sources of ions.

Providing the concentration information would be helpful for the reader in assessing this further. Maybe plots for the SI? I may be wrong but the PCA and isotope data alone do not provide what is needed to calculate this.

**AR:** Yes, the water interacts with the soil and vegetation, and this variability is picked up primarily by the first PC (we need to add more information of this in the revised version). The remaining variability in, e.g., Ca, cannot be explained by the interaction with the soil because according to the PCA this part of the variability in Ca plots together with Cl, and there is very little Cl-weathering (because very few minerals contain Cl). Furthermore, the mas-balance budget (Rydberg et al. 2023) suggests there is a rather large input of Cl (and other elements, e.g., Ca) with precipitation, and considering the massive amount of evapotranspiration we would argue that it actually is reasonable to couple the increase in Cl and Ca to the effect of evapotranspiration. This is a very dry environment and only part of the concentration increase in the lake can be explained by input from weathering.

From the reviewer comments it is obvious that we need to be much better at explaining this in the revised version (both in the methods and in the discussion). We hope that the revised version does a better job at describing this.

61. RC2 424-425:

water age means deeper footpaths/sources and thus more time interacting with soils and mineral particles as sources of ions.

**AR:** Yes, and we do discuss this further down in the text. However, from the lysimeter data we can also see that there isn't any clear pattern with depth, so in this particular case it is not possible to say that deeper flow paths always mean older water (and more weathering products). We have tried to put forward also the effect of soil particle interactions in the revised version.

62. RC2 429-430:

EXACTLY! So why are you claiming it is all from some sort of evaporation process? This is where I am now confused… you spent so much time saying it is all about this mythical evaporation and now say it might be from the deeper flow paths. I agree with the deeper flow paths source.

**AR:** Yes of course, the soil interaction is important (and this is represented by the first PC), and we never say otherwise. However, because elements that are present in the precipitation (and not supplied through weathering, e.g., Cl) behave differently compared to those "only" supplied via weathering and soil particle interaction (e.g., La) in the mass-balance budget (and plot on different components in the PCA) there must be an additional process involved. We have tried to explain this better in the revised version (see also earlier comment).

63. RC2 34-435:

again, look at that Cai paper and others. Increased river DOC from two sources/time periods: spring melt when there is a lot of movement of organic matter that degraded during winter. Then during summer rain events when more water flushed through slightly deeper flow paths into more organic matter sources.

**AR:** Yes, and we do not feel that what we write contradict this (in the revised version the Cai et al. 2008 reference is added here). When it comes to the activation of the second source of OM during summer (in deeper soil layers), it should be recognized that there is a rather limited flow of water in the catchment during the dry summer season. For example, there are no surface water flowing though this system during the summer (or unfrozen) season, and hence, the comparison to systems with permanent streams is a not always applicable.

64. RC2 511:

are these ice wedges or segregated ice? Any idea how old this ice is? If they are ice wedges they are predominantly comprised of snow melt. If they are segregated ice layers then they likely represent more of a mean annual precipitation. If it is old then regardless of their ice type/formation source waters the colder climate had lower d18O and dD values for those sources compared to today?

**AR:** We have not observed or sampled any ice-wedges or ice-lenses, but it is not unlikely that such features exist. The stable water isotope data for ground ice comes from water squeezed from thawed pieces of frozen ground (without segregated ice, just ice from small pore spaces). This was done in spring 2017, and the pieces of frozen ground came from the upper part of the active layer (just below the thaw horizon in early June), hence, the age of this water likely corresponds to the ages suggested by the particle tracing for the 0-25 cm layer. Hence, the difference in the d18O and dD should be related to recent processes in the catchment rather than an effect of climate. We have tried to clarify this in the revised version.

65.   RC2 513:

movement of meltwaters during the winter how? The surface is frozen (as you say earlier).

**AR:** From the weather station we see that there are occasions with temperatures above zero degrees and rain events also in winter, which drives the inflow of water to the lake in the hydrological model. And yes, the ground is frozen so presumably the flow occurs as overland flow (even if we have not observed this). The area does not have continuous snow cover (spatially) in winter, so it will behave a bit differently compared to completely snow-covered areas when it rains in winter (normally a large part of the rain would be retained in the snowpack). We have clarified this in the revised version.

66.   RC2 522:

consistent with patterns

**AR:** have been changed.

67.   RC2 536:

has also

**AR:** have been changed.

---

## Editor Decision (ED1)

Supplement comments on the manuscript "The coupling between hydrology, the development of the active layer and the chemical signature of surface water in a periglacial catchment in West Greenland" by Johan Rydberg, Emma Lindborg, Christian Bonde, Benjamin M. C. Fischer, Tobias Lindborg, and Ylva Sjöberg

Special issue: Northern hydrology in transition – impacts of a changing cryosphere on water resources, ecosystems, and humans (TC/HESS inter-journal SI)

To: Johan Rydberg, corresponding author

From: Svetlana Stuefer, handling editor

November 5, 2025

The authors have made significant progress in restructuring the manuscript. The reviewers still have concerns about typographical and grammatical errors. I encourage the authors to focus on improving readability and presentation quality and offer a few suggestions that could help with that.

1) Data use and availability: Consider improving the presentation of various datasets, their purpose, time periods, and data citations. I recommend adding a summary table (Supplemental File) that clearly lists each dataset (both used and produced), along with corresponding time periods, data availability (including data citation with DOI), and the purpose of each dataset for this study (e.g., field observations, model forcing, model calibration/validation, model outputs, statistical analysis). This table will enhance the readability of relevant sections (data, method, results, and discussion) and help ensure consistency. It will also assist others in replicating analyses and applying findings reported in the manuscript to future research.

2) Snowmelt period: The research objectives largely focus on the snowmelt period and snowmelt water; however, the data on snow water equivalent, snow ablation, and snowmelt rates are minimal in the current version of the paper. What are the implications of relying on snowfall data only? Multiple studies have shown that snowfall measurements are extremely problematic in windy treeless locations. Snow sublimation can be quite significant. Please clarify the use of winter precipitation and snow water equivalent data for estimating overland flow and direct runoff during snowmelt.

3) Conclusions: Consider framing your conclusions within the context of other northern hydrology studies and clearly highlight the unique contributions of your research in a broader context.

Specific comments:

Line 24: The term 'hydrological active season' – could you clarify what exactly is meant by this? Are you referring specifically to the month of September?

Line 44: Should it say "depends"?

Lines 116–119: Could you clarify question 3? What specific 'other important factors' are you referring to? Are you referring to factors such as groundwater, biological activity, or atmospheric deposition?

Line 179: How were snow ablation and the associated snowmelt rates represented in the model?

Line 222: How did you measure snow ablation during snowmelt? Did you conduct snow surveys?

Line 296–297: Was snow sublimation considered here?

Figure 2 and Figure 4: Figure 2B shows precipitation as snow and rain. Please add 'Snowmelt rates (mm/day)' to Figures 2B and 4C to indicate how much water leaves the snowpack daily.

Figure 3: Replace the comma with a period after Figure 3. The same comment applies to Figure 5.

Figure 3: Please use capital letters for the horizontal axis labels (Jan, Feb, Mar, etc) to ensure consistency with the labels in Figure 4.

---

## Author Response (AR2)

**Response to supplement comments on the manuscript "The coupling between hydrology, the development of the active layer and the chemical signature of surface water in a periglacial catchment in West Greenland"**

Johan Rydberg, Emma Lindborg, Christian Bonde, Benjamin M. C. Fischer, Tobias Lindborg, and Ylva Sjöberg

Dear Editor,

We like to thank both you and the two reviewers for the kind and constructive feedback. During the revision we have made a dedicated effort to increase the readability and make sure that text is grammatically correct. Below you will find detailed responses to each of the questions raised by you or the reviewers (in red). We hope that these responses, and the corresponding changes to the manuscript, will satisfactorily addresses all concerns. One thing that remains to be fixed is the links to the new Pangaea-datasets in SI-Table 1. The data have been uploaded to Pangaea, but we are still waiting to receive the doi-addresses for them. Hence, this information has to be added to the SI-Table at a later stage, we hope that this is not a problem.

Best regards,

Johan Rydberg

**Comments from Svetlana Stuefer, handling editor**

The authors have made significant progress in restructuring the manuscript. The reviewers still have concerns about typographical and grammatical errors. I encourage the authors to focus on improving readability and presentation quality and offer a few suggestions that could help with that.

1. Data use and availability: Consider improving the presentation of various datasets, their purpose, time periods, and data citations. I recommend adding a summary table (Supplemental File) that clearly lists each dataset (both used and produced), along with corresponding time periods, data availability (including data citation with DOI), and the purpose of each dataset for this study (e.g., field observations, model forcing, model calibration/validation, model outputs, statistical analysis). This table will enhance the readability of relevant sections (data, method, results, and discussion) and help ensure consistency. It will also assist others in replicating analyses and applying findings reported in the manuscript to future research.

AR: We have added such a table to the supplementary information

2. Snowmelt period: The research objectives largely focus on the snowmelt period and snowmelt water; however, the data on snow water equivalent, snow ablation, and snowmelt rates are minimal in the current version of the paper. What are the implications of relying on snowfall data only? Multiple studies have shown that snowfall measurements are extremely problematic in windy treeless locations. Snow sublimation can be quite significant. Please clarify the use of winter precipitation and snow water equivalent data for estimating overland flow and direct runoff during snowmelt.

AR: Yes, sublimation and spatial heterogeneity is tricky to constrain. We have added more information regarding how this was dealt with in the modelling. See further below

3. Conclusions:Consider framing your conclusions within the context of other northern hydrology studies and clearly highlight the unique contributions of your research in a broader context.

AR: We have considered the suggestion, but in our view the main purpose of the conclusions is to present the main findings of this particular study in a nutshell. A key feature of a nutshell is that it should be condensed and short, and hence, we have not added any additional parts to our conclusions. Instead, the unique aspects of this study as well as how it compares to a broader context can – hopefully – be found in the Abstract, Introduction and Discussion. We feel that this should be sufficient.

**Specific comments:**

Line 24: The term 'hydrological active season' – could you clarify what exactly is meant by this? Are you referring specifically to the month of September?
AR: Yes and no, the intention here is not to mention a specific month (because it varies between years). We have now changed this to "...the end of the thawed season."

Line 44: Should it say "depends"?
AR: Yes, this was changed

Lines 116–119: Could you clarify question 3? What specific 'other important factors' are you referring to? Are you referring to factors such as groundwater, biological activity, or atmospheric deposition?
AR: We have rewritten the question to make it clear what we refer to with "other important factors"

Line 179: How were snow ablation and the associated snowmelt rates represented in the model?
AR: The model handles sublimation in the same way as it handles evaporation, that is, it is estimated based on the meteorological observations. We have added more information on this in the method section.

Line 222: How did you measure snow ablation during snowmelt? Did you conduct snow surveys?
AR: We didn't. On a catchment scale snow ablation was assumed to be equal to the accumulation of snow drifts, i.e., no net loss or gain of any water to the water balance. Of course, for a finer scale than the catchment or sub-catchment scale this is not a valid assumption. We have added more information regarding this in the methods

Line 296–297: Was snow sublimation considered here?
AR: Yes, with the addition of more information regarding sublimation in the methods and the reformulation here it should hopefully be clear that evapotranspiration includes sublimation.

Figure 2 and Figure 4: Figure 2B shows precipitation as snow and rain. Please add 'Snowmelt rates (mm/day)' to Figures 2B and 4C to indicate how much water leaves the snowpack daily.
AR: Adding the snowmelt rate is not as straight forward as it might seem at first glance. Yes, the snow melts and as this happens water is released in the hydrological model (and it is then

partitioned in the same way as water entering as precipitation). However, after it has melted it might (depending on the conditions) refreeze as ice on the ground surface or further down in the catchment. As the temperature then increases this ice will melt again (but this will not count as snowmelt). The problem is then to separate the different types of melting water based on if it comes from snow, ice or something else. Just looking at the initial melting of snow will underestimate the release of water in the catchment (or make it appear as if it produces runoff earlier), and including all types of melting water will just reproduce the total runoff during spring (which is driven by the melting of snow and ice in the catchment). Hence, we opted for including the accumulation of water in the form of snow in the catchment to the graphs in figure 2 and 4. In our opinion this better shows the illustrate the dynamics in the accumulation and release of water from the "snow pool" (rather than trying to separate the different types of melting water from each other).

Figure 3: Replace the comma with a period after Figure 3. The same comment applies to Figure 5.
AR: Fixed

Figure 3: Please use capital letters for the horizontal axis labels (Jan, Feb, Mar, etc) to ensure consistency with the labels in Figure 4.
AR: Fixed

**Comments from Reviewer 1**

No comments, publish as is

**Comments from Reviewer 2**

**General comments:**

This version is greatly improved. All of my comments/suggestions have been adequately addressed. I applaud the hard work addressing the comments from the two reviewers. The revised text still has a lot of typographical and grammatical errors that hopefully can be fixed by the Editorial process (?). I suggest this work is ready for publication. A few specific comments are provided below.
AR: We thank the reviewer for the kind comments. Regarding the language one reason for the typographical and grammacial errors might be that the track changes version of the resubmitted manuscript unfortunately hadn't gone through the fiinal langauge check (we are sorry for this). Anyways, we have gone through the text once more to correct any remaining issues. We have, howver, gone through the text and tried to make it as readable as we can (and tried to correct all grammatial errors)

Comments keyed to the text:
248-252: without permafrost there is groundwater flow as well
AR: We have refrased this to make it clear that groundwater is present also without permafrost

414: any relationship
AR: Changed

418: or do our
AR: Changed

589: I realize Reviewer 1 suggested a reorganization but it is not clear to me why the modeling comes before the measurements. The data used to do the modeling should come before the modeling itself.'

AR: That is a valid point, however, in this case we argue that it still is more logical to put the modelling first. First, the modelling includes a longer time period than most of the measurements. We feel that the reader will benefit from first being presented with this longer time period and the seasonality of the system, before going into more detail with the samples from the snowmelt season. Second, the modelling builds on previously published data, and apart from the "Meteorology and ground temperature" none of the data presented under "Sample collection…" has been used for the modelling. Third, the modelling data is used to assess the chemical and isotopic data, and it would be difficult to write about the results from the sampling without referring forward to results from the modeling (and we like to retain the same order in both the methods and results).

775: "found" is not needed. Maybe say "located" instead?
AR: Changed

1260: suggests that
AR: Changed

1485: indicates
AR: Changed

1597: dominate runoff
AR: Changed

1672: The effects of fast shallow flow paths…
AR: Changed

1673: that have
AR: Changed

1678: moves through
AR: Changed